

# TREMOL: A stochastic rupture earthquake code based on the fiber bundle model. Application to Mexican subduction earthquakes.

Marisol Monterrubio-Velasco[1], Quetzalcóatl Rodríguez-Pérez[2,3], Francisco Ramón Zúñiga[2],
Doreen Scholz[4], Armando Aguilar-Meléndez[1,5], and Josep de la Puente[1]

[1]Barcelona Supercomputing Center, Jordi Girona 29, C.P. 08034, Barcelona (Spain)
[2]Centro de Geociencias, Universidad Nacional Autónoma de México, Juriquilla, Querétaro, 76230, México
[3]Consejo Nacional de Ciencia y Tecnología, Mexico City, 03940, Mexico.
[4]Fugro Germany Land GmbH, Wolfener Str. 36 U, 12681 Berlin, Germany
[5]Facultad de Ingeniería Civil, Universidad Veracruzana, Poza Rica, Veracruz, 93390, México

**Correspondence:** Marisol Monterrubio-Velasco (marisol.monterrubio@bsc.es)

**Abstract.** In general terms, earthquakes are the result of brittle failure within the heterogeneous crust of the Earth. However, the rupture process of a heterogeneous material is a complex physical problem which is difficult to model deterministically due to the numerous parameters and physical conditions, which are largely unknown. Considering the variability within the parametrization, it is necessary to analyze earthquakes by means of different approaches. Computational physics may offer
alternative ways to study brittle rock failure by generating synthetic seismic data based on physical and statistical models, and by the use of only few free parameters. The Fiber Bundle model (FBM) is a discrete element model, which is able to describe complex rupture processes in heterogeneous materials. In this article, we present a computer code called *stochasTic Rupture Earthquake MOdeL*, TREMOL. This code is based on the principle of the FBM to investigate the rupture process of asperities on the earthquake rupture surface. In order to validate TREMOL, we carried out a parametric study at first to identify
the best parameter configuration while minimizing computational efforts. As test cases, we applied the final configuration to 10 Mexican subduction zone earthquakes in order to compare the synthetic results by TREMOL with real data. According to our results, TREMOL is able to model the rupture of an asperity that is defined essentially by two basic dimensions: (1) the size of the fault plane, and (2) the size of the maximum asperity within the fault plane. Based on this data, and few additional parameters, TREMOL is able to generate numerous earthquakes as well as a maximum magnitude for different scenarios within
a reasonable error range. The simulated earthquakes magnitudes are of the same order as the real earthquakes. Thus, TREMOL can be used to analyze the behavior of a single asperity or a group of asperities since TREMOL considers the maximum magnitude occurring on a fault plane as a function of the size of the asperity. TREMOL is a simple, and flexible model which allows its users to investigate the role of the initial stress configuration, and the dimensions and material properties of seismic asperities. Although various assumptions and simplifications are included in the model, we show that TREMOL can be a
powerful tool which can deliver promising new insights into earthquake rupture processes.

*Copyright statement.* TEXT





# 1   Introduction

Rupture models of large earthquakes suggest significant heterogeneity in slip and moment release over the fault plane, (e.g., Aochi and Ide, 2011). In order to characterize the seismic source rupture complexity, two main models have been proposed: the asperity model (Kanamori and Stewart, 1978), and the barrier model (Das and Aki, 1977). Asperities are defined as regions on the fault rupture plane that have larger slip and strength in comparison to the average values on the fault plane (Somerville et al., 1999). Asperities also have larger stress drop than the background area (Madariaga, 1979; Das and Kostrov, 1986). Understanding the physical features in the fault zone that produce these high-slip regions is still a challenge.

The most common method for studying seismic asperities is waveform slip inversion. However, information obtained from this method is highly variable due to the inherent nature of the inversion process (see review in Scholz (2018)). The slip inversion results depend on the type of data (such as strong ground motion, geodetic and/or seismic data at different distances) and the inversion technique used. Somerville et al. (1999) used average slip to define asperities. In their criterion, asperities include fault elements where slip is 1.5 times or more larger than the average slip. By using this criterion, it is possible to estimate the asperity area from a finite-fault slip model. Considering the stress drop for a circular crack model ($\Delta\sigma$) (Eshelby, 1957), the stress drop on an asperity ($\Delta\sigma_a$) can be estimated as $\Delta\sigma_a = (A_{\mathrm{eff}}/A_{\mathrm{a}})\Delta\sigma$, where $A_{\mathrm{eff}}$ and $A_{\mathrm{a}}$ are the rupture effective area, and the asperity area, respectively (Madariaga, 1979). The $A_{\mathrm{eff}}/A_{\mathrm{a}}$ factor (or its reciprocal value) depends on different features with the most relevant one being the type of earthquake. For example, Somerville et al. (1999) found that on average the total area covered by asperities represents 22% of the total rupture area for inland crustal events. Murotani et al. (2008) showed that $A_{\mathrm{a}}/A_{\mathrm{eff}}$ is approximately equal to 20% for plate-boundary events. Similarly, for subduction events, the value of $A_{\mathrm{a}}/A_{\mathrm{eff}}$ is approximately equal to 25% (Somerville et al., 2002; Rodríguez-Pérez and Ottemöller, 2013). The previous average values were determined considering values that range from 0.09 to 0.35. This last condition means, for instance, that the reciprocal fraction $A_{\mathrm{a}}/A_{\mathrm{eff}}$ can deviate from these average values as well (for example 0.09 to 0.35 for the proportions mentioned above), which leads to great stress contrasts (factors of 2.8 to 11) (Iwata and Asano, 2011; Murotani et al., 2008). Mai et al. (2005) proposed another definition of asperities based on the maximum displacement, $D_{\max}$. They defined "large-slip" and "very-large-slip" asperities as regions where the slip $D$ lies between $0.33D_{\max} \leq D < 0.66D_{\max}$, and $0.66D_{\max} \leq D$, respectively. They found that approximately 28% of the rupture plane is occupied by large-slip asperities, whereas very-large-slip areas constitute only 7% of the fault plane. Furthermore, different authors agree that the rupture area of the asperity scales with the seismic magnitude (Somerville et al., 1999; Murotani et al., 2008; Iwata and Asano, 2011; Rodríguez-Pérez and Ottemöller, 2013, among others). The estimation of seismic magnitude is an essential feature for characterizing the energy of an earthquake. In fact, an accurate magnitude estimation is indispensable to do both deterministic and probabilistic seismic hazard assessments. Earthquakes are the most relevant example of self-organized criticality (SOC) (Bak and Tang, 1989; Olami et al., 1992). The concept of SOC can be visualized by imagining a natural system in a marginally stable state, where phases of instability may occur which place the system back into a meta-stable state (Barriere and Turcotte, 1994). A popular model representing this process was proposed by Bak and Tang (1989) and is well-known as the "sand pile model". Some models have been proposed to explain the statistical behavior of earthquakes patterns based on the SOC concept, e.g. Caruso et al.



(2007), Barriere and Turcotte (1994), Olami et al. (1992), Bak and Tang (1989). The failure properties of solids have been modeled by simple discrete element models, which are based on the SOC framework. The Fiber Bundle Model, FBM, is one of those models which has been used to reproduce many basic properties of the failure dynamic within solids (Chakrabarti and Benguigui, 1997). Additionally, the FBM has been successfully applied to studies of brittle failure of rocks (Hansen et al.,

2015; Monterrubio et al., 2015; Turcotte and Glasscoe, 2004; Moreno et al., 2001).

## 2   The Fiber Bundle Model

The FBM is a numerical approach to study the rupture process of heterogeneous materials which was originally introduced by Peirce (1926). Over the years the FBM has been widely used to study failure in a wide range of heterogeneous materials

(Hansen et al., 2015; Pradhan and Chakrabarti, 2003). Regardless of the specific FBM type, there are three basic assumptions that all FBMs have in common (Daniels, 1945; Andersen et al., 1997; Kloster et al., 1997; Vázquez-Prada et al., 1999; Phoenix and Beyerlein, 2000; Pradhan et al., 2010; Monterrubio-Velasco et al., 2017):

1. A discrete set of cells (or fibers) which are defined on a $d-$dimensional lattice. In seismology, the bundle can represent a

fault system, or seismic source where each fiber is a section of the fault plane (Moreno et al., 2001), or individual faults (Lee and Sornette, 2000).

2. A probability distribution that defines the inner properties of each cell (fiber), such as lifetime, or stress distribution.

3. A load-transfer rule which determines how the load is distributed from the ruptured cell to its neighbor cells. The most common load-transfer rules are: (a) Equal Load Sharing (ELS) in which the distributed load is equally shared to the other cells within the material or bundle; and (b) Local Load Sharing (LLS) where the transferred load is only shared with the nearest neighbors.

TREMOL is based on the probablistic formulation of the FBM, with the failure rate of a set of fibers given by Eq. 1 (Gómez

et al., 1998; Moral et al., 2001).

$$\frac{dU(t)}{dt} = -U(t)K(\sigma(t)),\tag{1}$$

where $U(t)$ is the number of fibers that remain unbroken at time $t$. The hazard rate $K(\sigma(t))$ is a function of the fiber stress $\sigma(t)$. Experimental results show that the hazard rate of materials under constant load can be well described by the Weibull





probability distribution function. This behavior can be summarized in Eq. 2 (Coleman, 1958; Phoenix, 1978; Phoenix and Tierney, 1983; Vázquez-Prada et al., 1999; Moreno et al., 2001).

$$K(\sigma(t)) = \nu_0 \left( \frac{\sigma(t)}{\sigma_0} \right)^\rho \tag{2}$$

where $\nu_0$ is the reference hazard rate, and $\sigma_0$ the reference stress. The Weibull exponent, $\rho$, quantifies the non-linearity
(Yewande et al., 2003). If $\sigma_0 = \nu_0 = 1$, the expression in Eq. 2 can be simplified to $K(\sigma(t)) = \sigma(t)^\rho$. From the probabilistic formulation, two equations arise (Eq. 3 and Eq. 4), which are applied in our algorithm to define the system dynamics. The details of these two equations are mentioned below.

   a) Gómez et al. (1998), and Moral et al. (2001), developed a relation to compute the expected rupture time [dimensionless] of the fibers following Eqs. 1, and 2. This expected rupture time interval is defined as $\delta_k$ (Eq. 3), and can be applied to
any load transfer rule,

$$\delta_k = \frac{1}{\displaystyle\sum_{i=1}^{N} \sigma_i^\rho(t)} , \tag{3}$$

   where $N$ is the total number of cells, and $\sigma_i$ is the load in the $i^{th}$ cell. The dimensionless cumulative time, $T$, is the sum of $\delta_k$.

   b) The failure probability, $F_i$, which is a function of the load $\sigma_i$ in each cell is (Moreno et al., 2001),

$$F_i = \delta_k \sigma_i^\rho(t) . \tag{4}$$

   The dynamic values $\delta_k$, and $F_i$ are updated with each time-step due to rupture processes, and the resulting load transfer.

   A suitable FBM algorithm to simulate earthquakes should consider a complex stress field, physical properties of materials, stress transfer between faults (at short and long distances), and dissipative effects. Using the FBM we assume that earthquakes can be considered as analog to characteristic brittle rupture of a heterogeneous material (Kun et al., 2006a, b).
The previous basic concepts about the FBM were considered for the development of the TREMOL code, with the purpose of modeling the behavior of seismic asperities. In the next section, we describe details of this code.

## 3   The TREMOL code

Since the main objective of TREMOL is to simulate the rupture process of seismic asperities based on the principles of the FBM, we model two materials with different mechanical properties interacting with each other.
In order to introduce the features of TREMOL we describe three main stages during the application of TREMOL.

(c) Author(s) 2019. CC BY 4.0 License.

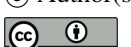



1. Pre-processing

    In this stage we have to assign the following input data:

    – the size of the fault plane,

    – the size of the maximum asperity within the fault plane,

– other parameters (they are described with detail in this document).

2. Processing

    TREMOL uses the data of the pre-processing stage to carry out the FBM algorithm, and applying Eqs. 3 and 4 is computed the rupture process in the fault plane studied. The asperity size of each earthquake is used by TREMOL to compute also the magnitude of each synthetic earthquake.

3. Post-processing

    In this stage, TREMOL summarizes the results that is computed in processing stage and computes the equivalent rupture area $[km^2]$. In general, TREMOL output generates a synthetic catalogue of earthquakes, which consists of the following:

    – total number of earthquakes that can occur in the the fault plane studied,

    – size of the asperity of each earthquake,

– magnitude of each earthquake.

In the next sections we describe with more detail each one of the three main stages during the application of TREMOL. An overview of the entire simulation process is shown in Fig. 1.

### 3.1 Pre-processing: Input data and initial conditions

In TREMOL, a fault plane is modeled as a rectangle ($\Omega$), and it is divided into $N_x \times N_y$ cells. Each cell is defined by its

position (i, j), where $i \in [1, ..., N_x]$, and $j \in [1, ..., N_y]$. In the fault plane $\Omega$ earthquakes can occur with different magnitudes. Additionally, it is possible to assign to each fault plane an asperity region ($R_{Asp}$).

To define each fault plane ($\Omega$), and its respective asperity region ($R_{Asp}$) is necessary to assign specific properties to their cells. Particularly, it is necessary to define three properties (or values) for each cell of $\Omega$ and $R_{Asp}$: a load $\sigma(i,j)$, a strength value $\gamma(i,j)$, and a load-transfer value $\pi(i,j)$.

– The load $\sigma(i,j)$. At the beginning of each realization, TREMOL assigns randomly a value of the load $\sigma(i,j)$ to each cell of $\Omega$ using a uniform distribution function ($0 < \sigma(i,j) < 1$). Without loosing generality, this assumption simulates a heterogeneous stress field. Moreover, a load threshold $\sigma_{th} = 1$ is defined to identify the amount of load required to break a cell (Moreno et al., 2001). In summary, at the end of this step any cell of $\Omega$ must have a value of load between 0 and 1.

    – The strength value $\gamma(i,j)$. This parameter represents an analogy to the concept of hardness or strength. In our model, the

algorithm will find it difficult to break a cell if this cell has a value $\gamma > 1$ since the strength threshold before failure is set



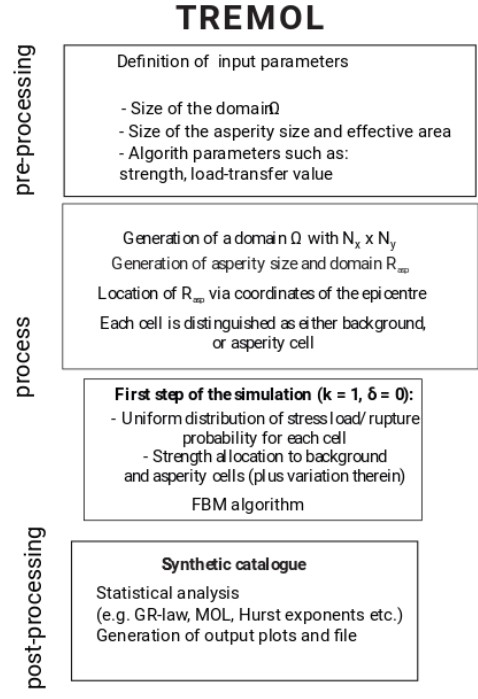

**Figure 1.** TREMOL flowchart. At the beginning (pre-process) the algorithm initiates a domain $\Omega$ with $N_x \times N_y$ cells where every cells is either part of an asperity or of the background or fault plane. Afterwards (first time-step, $k = 1$) a uniform distribution allocates a random stress load and rupture probability to all cells. In addition, asperity cells obtain a random strength value from a uniform distribution. At next (time-step $k \geq 2$) the failure process starts following the FBM algorithm. After every failure the stress of the broken cell is redistributed via the LLS rule and the number of time-steps ($k$) increases by 1 until the target number of time-steps is reached. In this case the simulation stops. In the end, all information about the entire failure process are saved in a database/synthetic catalogue which can be used for statistical analysis. Further details about the algorithm are given in section 3.

as $\gamma_{\text{th}} = 1$ (see a detailed explanation in Subsection 3.2). As a result, a strength $\gamma > 1$ may simulate a hard material which needs to be weakened before it can fail. This process can be regarded as a simile to material fatigue or creep failure. The strength value for all cells in $R_{\text{Asp}}$, namely $\gamma_{\text{Asp}}$, is chosen in a discrete interval of $\mathcal{U}_D = [\gamma_{\text{Ref}} - 1, \gamma_{\text{Ref}} + 1]$, where $\mathcal{U}_D$ is an integer uniformly distributed, and $\gamma_{\text{Ref}}$ is an assigned reference value.

– The load-transfer value $\pi(i,j)$. This parameter represents the percentage of load that can be distributed from a ruptured cell to its neighbors. In this study, the load in the ruptured cell is called $\sigma_F(i,j)$. TREMOL uses a local load rule considering the eight nearest neighbors. According to previous studies, such as Monterrubio-Velasco et al. (2017), TREMOL redistributes the majority of load to the four orthogonal neighbors. The load that is transferred to these orthogonal neigh-





bors is called $\sigma_O$ and it is defined according to Eq. 5:

$$\sigma_O(i,j) = \frac{0.98\sigma_F(i,j)\pi_F(i,j)}{4}, \tag{5}$$

where $\pi_F$ is the load-transfer value of the failed cell. Additionally, a small proportion of the load is transferred to the four diagonal neighbors. The value of this load is called $\sigma_D(i,j)$, and it is defined according to Eq. 6:

$$\sigma_D(i,j) = \frac{0.02\sigma_F(i,j)\pi_F(i,j)}{4}. \tag{6}$$

Fig. 2 represents in a schematic way the load distribution process from the failed cell, $\sigma_F(i,j)$ (in red color), to its nearest neighbors.

In order to differentiate the parameters of the asperity with the rest of the fault plane $\Omega$, we define $\pi_{Asp}(i,j)$ and $\gamma_{Asp}(i,j)$ that refer only to the cells in $R_{Asp}$. For the rest of the fault plane $\Omega$, we are using the same parameters defined previously $\pi(i,j)$
as well as $\gamma(i,j)$.

Fig. 3(a) shows an example of the randomly distributed initial load throughout the fault plane. Fig. 3(b) displays an example of differences between the strength of the asperity and the rest of the fault plane.

### 3.2 Main computational processes

Once the initial information for the entire domain $\Omega$ is defined, the core algorithm of TREMOL will realize a transfer, accu-
15 mulation and rupture process. While the cells interact with each other, there are two basic failure processes depending on the load of the cell in comparison with the threshold load (Moreno et al., 2001):

- *Normal event*: If all cells within the system have a load $\sigma(i,j) < \sigma_{th}$, a *normal-event* is generated, and the cell that will fail is randomly chosen considering the individual failure probability of each cell, $F(i,j)$ (Eq. 4).

- *Avalanche event*: If one or more cells have a load value $\sigma(i,j) \geq \sigma_{th}$, an *avalanche-event* is generated, and the cell
that fails is the one with the greatest $\sigma(i,j)$ value.

Due to the integrated strength property some extra rules for rupture are necessary. The requirement for failure is $\gamma(i,j) = 1$. On the other hand, if a cell with $\gamma(i,j) > 1$ is chosen, its strength is reduced by one unit. This strength condition enables us to simulate material weakness or fatigue during the load transfer process. Additionally, this condition offers the possibility to produce large load accumulations locally which are more likely to generate larger ruptures.
When a cell within the $R_{Asp}$ breaks it becomes inactive until the end of the simulation which means it cannot receive any further load. The large load concentration within the asperity usually produces a very short time interval (Eq. 3), and physically there is not enough time available to re-load the stress on an asperity right after its rupture. On the contrary, a cell outside of the asperity region remains active after its failure but its load drops to zero. The simulation ends when all the cells within the asperity have become inactive.




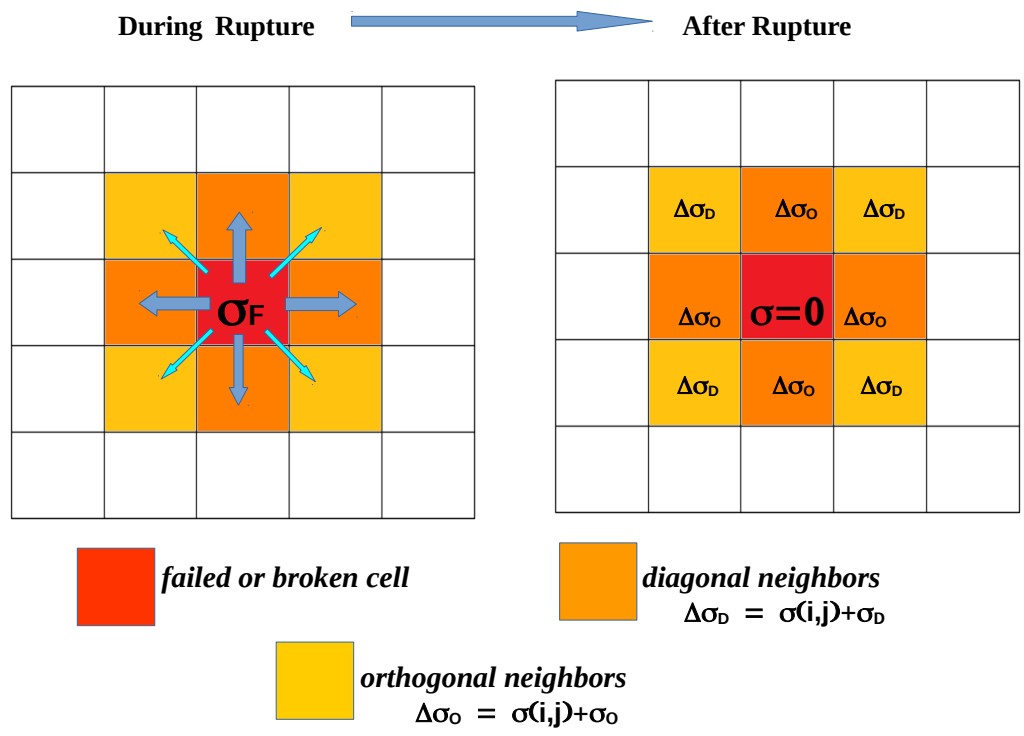

**Figure 2.** Schematic representation of the considered local load rule. The broken cell with load, $\sigma_\mathrm{F}$, distributes the largest load fraction, $\sigma_\mathrm{O}$ (Eq. 5), to its four orthogonal neighbor cells. The remaining load, $\sigma_\mathrm{D}$ (Eq. 6), is transferred to its four diagonal neighbor cells. Afterwards, the load of the broken cell drops to zero, $\sigma = 0$. Asperity cells cannot receive any new load.

### 3.3 Output data and post processing

After every execution TREMOL outputs a catalog where the position (x,y) of the failed cell, the rupture time (Eq. 12), the *Avalanche* or *Normal* event identification, the mean load, and many other values are saved for each time-step. We cluster avalanche events considering the time and space criterion. Assuming $a_{i-1} = (x_{i-1}, y_{i-1})$ and $a_i = (x_i, y_i)$ being both two

5 consecutive *avalanche-events* generated in chronological order. If their euclidean distance $\Delta r_i \leq r_{th}$, (where $r_{th} = \sqrt[2]{2}$), then $a_i$ and $a_{i-1}$ will belong to the same cluster. This clustering algorithm is applied to all generated *Avalanche events*. Lastly, we extract a new catalog that shows the size of each cluster, the position of the first element of each cluster, related to the nucleation point, and the time when it was initiated. This database is our simulated seismic catalog. Note that the cluster size is given in non-dimensional units. However, we use an equivalence between $\Omega$ and an effective area $A_{\mathrm{eff}}$ in order to

10 obtain a physical rupture area. Finally, each cell can represent an area in $\mathrm{km}^2$. This step is necessary in order to compute an equivalent magnitude, which is comparable with real earthquake magnitudes. For this purpose, we use three magnitude-





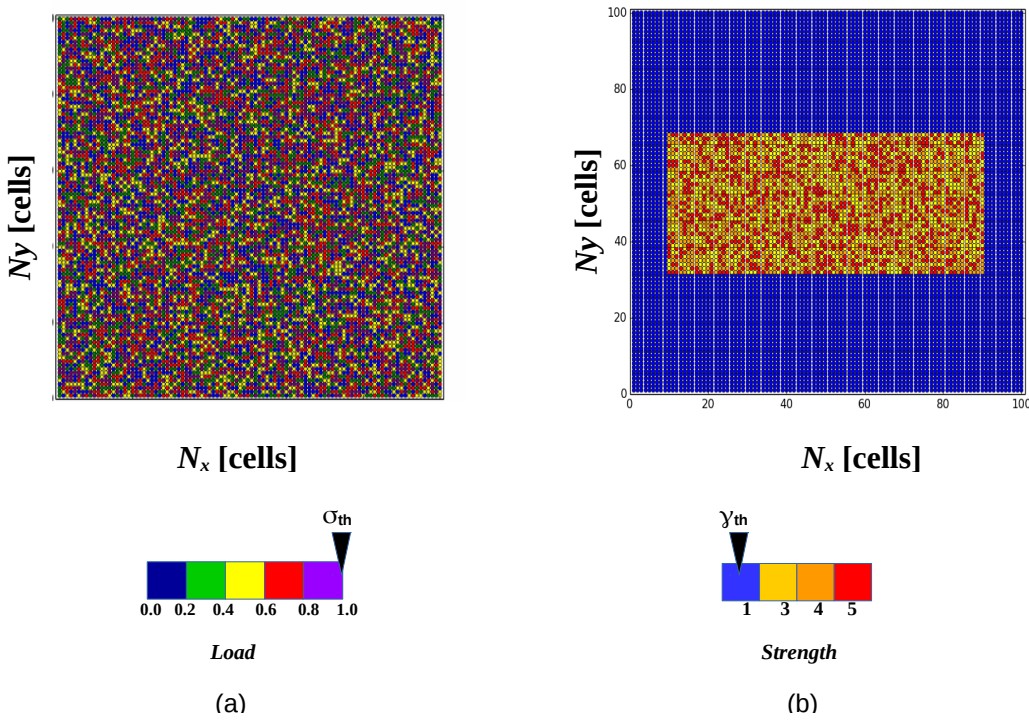

**Figure 3.** (a) Spatial distribution of the random initial loads $\sigma(i,j)$. $R_{\mathrm{Asp}}$ represents a rectangular fault plane of $N_x = N_y = 100$ cells. The color bar indicates the load and the threshold load of $\sigma_{\mathrm{th}} = 1$. (b) Spatial distribution of the strength $\gamma(i,j)$. Two main regions can be distinguished in this figure: 1) the asperity region defined as the inner rectangle, and 2) a background area or fault plane. While the asperity contains strength values in the range of 3 to 5, the rest of the fault plane has a strength value of 1.

area-relations. In particular, we use Eqs. 7, 8, and Eq. 9, obtained by Rodríguez-Pérez and Ottemöller (2013) for Mexican subduction earthquakes:

$$\log_{10} A_{\mathrm{a}} = -4.393 + 0.991 M_w, \tag{7}$$

$$\log_{10} A_{\mathrm{a}} = -5.518 + 1.137 M_w, \tag{8}$$

5   $$\log_{10} A_{\mathrm{a}} = -6.013 + 1.146 M_w, \tag{9}$$

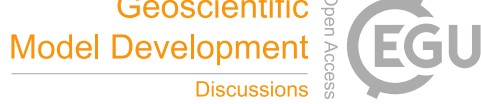

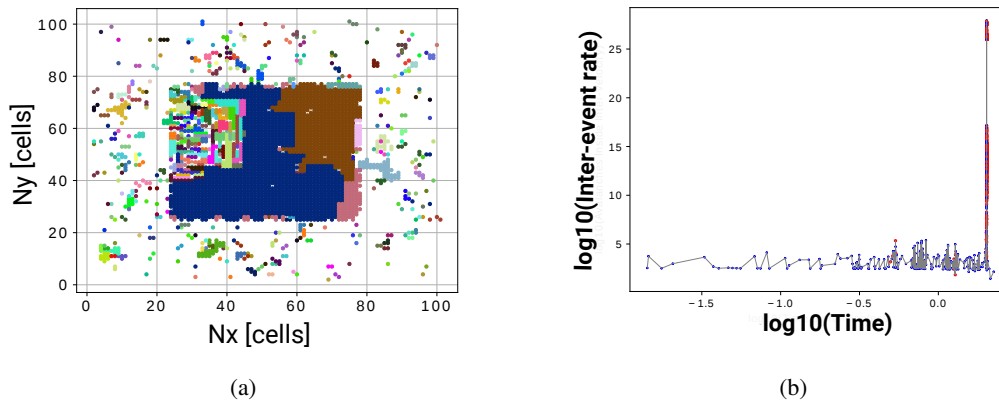

|  |  |
|:---:|:---:|
| (a) | (b) |

**Figure 4.** Results of one realization by TREMOL. (a) The spatial distribution of avalanches. Patches of the same color indicate one temporal-consecutive Avalanche cluster (synthetic earthquake). (b) Logarithmic representation of the inter-event rate with time. The red dots represent the inter-event rate when the asperity rupture occurs. The blue dots indicate foreshocks.

where $A_{\mathrm{a}}$ is the asperity area [km$^2$]. Eq. 7 was obtained from asperities defined by the average displacement criterion (Somerville et al., 1999). Eqs 8 and 9 were computed from asperities defined by the maximum displacement criterion for a large asperity and a very large asperity, respectively (Mai et al., 2005).

Furthermore, we define the inter-event rate $\Delta\nu_k$ to measure analog to the rupture velocity, as,

$$\Delta\nu_k = \frac{\Delta r_k}{\Delta\delta_k}, \tag{10}$$

where $\Delta r_k$ is the inter-event euclidean distance between the $k-event$ located at $(x_k, y_k)$, and $k-1$ event in $(x_{k-1}, y_{k-1})$.

$$\Delta r_k = \sqrt{(x_k - x_{i-k})^2 + (y_k - y_{i-k})^2}. \tag{11}$$

And $\Delta\delta_k$ is the inter-event time computed following

$$\Delta\delta_k = \delta_k - \delta_{k-1}, \tag{12}$$

where $\delta_k$ is given by Eq. 3. Figure 4a shows an example of the final spatial distribution of rupture clusters for a particular example. Each cluster is indicated by the same color and represents a simulated earthquake. Figure 4b shows the related inter-event rate. As one can see, the inter-event rate largely increases when the asperity rupture occurs.

In the post-processing step we additionally computed the rupture duration of the largest simulated quake, $D_{\mathrm{Aval}}$, using the rupture velocity, and the effective fault dimensions obtained from finite-fault models (Table 5).



**Table 1.** TREMOL pre-processing: Input parameters and their definition.

| Parameter | Definition |
|-----------|------------|
| $N_{\mathrm{cell}}$ | number of cells in $\Omega$ |
| $\pi_{\mathrm{asp}}$ | percentage of transferred load to neighbor cells in the asperity domain |
| $\gamma_{\mathrm{asp}}$ | strength at each (i,j) cell in the asperity domain |
| $S_{\mathrm{a-Asp}}$ | ratio of asperity area computed by TREMOL |
| $S_{\mathrm{a}}$ | ratio of asperity area computed by finite fault model |
| $A_{\mathrm{eff}}$ | effective area $[\mathrm{km}^2]$ computed by finite fault model |
| $A_{\mathrm{a}}$ | asperity area $[\mathrm{km}^2]$ computed by finite fault model |

We used Eq. 13 (Geller, 1976) to compute $D_{\mathrm{Aval}}$

$$D_{\mathrm{Aval}} = \frac{L_{\mathrm{Max}}}{V_{\mathrm{r}}} + \frac{16\sqrt{W_{\mathrm{Max}} \times L_{\mathrm{Max}}}}{7\pi^{3/2}\beta}, \tag{13}$$

where

$$\beta \approx \frac{V_{\mathrm{r}}}{0.72}. \tag{14}$$

5  Using these considerations, we can assign a physical unit of time [seconds] to the largest simulated earthquake, $A_{\mathrm{syn}} = L_{\mathrm{Max}} \times W_{\mathrm{Max}}$.

The flowchart in Fig.1 and the pseudo-codes 1, 2, and 3 summarize the algorithm of TREMOL. A summary of all required parameters to execute the TREMOL code are shown in Table 1.



---

**Algorithm 1** Basic FBM. Main algorithm of TREMOL which applies the Algorithm 3 in regards to the **initial conditions** procedure and to the **rupture** procedure, respectively.

---

$k = 0;\ n_A = 0;\ T_0 = 0$
$\delta_0 = \left( \sum_{i,j} \sigma(i,j)^\rho \right)^{-1}$ (Eq. 3)
**while** $k < k_{max}$ **do**
    $k = k + 1$
    **for all** $(i,j) \in \Omega$ **do**
        $F(i,j) = \sigma(i,j)^\rho \delta_k$ (Eq. 4)
    **end for**
    $(l,m) = \{(i,j) \in \Omega \mid \sigma(i,j) = \max(\sigma)\}$
    $(l*,m*) = \textbf{selection}(l,m)$ (Algorithm 2)
    **if** $\sigma(l*,m*) > \sigma_{\text{th}}$ **then**
        $n_A = n_A + 1$
        **rupture**$(l,m)$
        **if** $n_A = 1$ **then**
            $S(n_A) = 0$
        **else**
            $S(n_A) = S(n_A) + 1$
        **end if**
        $t(n_A) = T_k;\ S(n_A) = 0;\ E_x(n_A) = l;\ E_y(n_A) = m$
    **else**
        **if** $n_A \neq 0$ **then**
            $N_A = n_A$
            $S(N_A) = S(n_A)$
            $T(N_A) = t(n_A = 1)$
            $n_A = 0;\ S(n_A) = 0$
        **end if**
        find $(p,q)$ sample of $F(i,j)$
        **rupture**$(p,q)$
    **end if**
**end while**

---





---

**Algorithm 2** Identification of the next rupture cell. The algorithm of TREMOL which identifies the next cell whose percentage of load will be transferred.

---

$\quad$ **selection**$(l,m)$

$\quad$ **if** $\sigma(l,m) > \sigma_{th}$ **then**

$\quad\quad$ **if** $\gamma(l,m) = 1$ **then**

$\quad\quad\quad$ **return**$(l,m)$

$\quad\quad$ **else**

$\quad\quad\quad$ $\gamma(l,m) = \gamma(l,m) - 1$

$\quad\quad\quad$ $(l',m') = \{(i,j) \in \Omega \mid \gamma(i,j) = 1 \text{ and } \sigma(i,j) = \max(\sigma)\}$

$\quad\quad\quad$ **return**$(l',m')$

$\quad\quad$ **end if**

$\quad$ **else**

$\quad\quad$ find $(l^*,m^*)$ sample of $F(i,j)$

$\quad\quad$ **if** $\gamma(l^*,m^*) = 1$ **then**

$\quad\quad\quad$ **return**$(l^*,m^*)$

$\quad\quad$ **else**

$\quad\quad\quad$ $\gamma(l^*,m^*) = \gamma(l^*,m^*) - 1$

$\quad\quad\quad$ $(l^*,m^*) = \{(i,j) \in \Omega \mid \gamma(i,j) = 1 \text{ and } \sigma(i,j) = \max(\sigma)\}$

$\quad\quad\quad$ **return**$(l^*,m^*)$

$\quad\quad$ **end if**

$\quad$ **end if**

---

**Algorithm 3** Failure of a cell. The algorithm of TREMOL which computes the failure process in the model.

---

$\quad$ **rupture**$(p,q)$

$\quad$ $\sigma(p,q) = \pi(p,q)\,\sigma(p,q)$

$\quad$ **for** $(r,s) \in \{(1,0),(0,1),(-1,0),(0,-1)\}$ **do**

$\quad\quad$ $\sigma(p+r,s+q) = \sigma(p+r,s+q) + [\sigma_N \sigma(p,q)]$

$\quad$ **end for**

$\quad$ **for** $(r,s) \in \{(1,1),(1,-1),(-1,1),(-1,-1)\}$ **do**

$\quad\quad$ $\sigma(p+r,s+q) = \sigma(p+r,s+q) + [\sigma_D \sigma(p,q)]$

$\quad$ **end for**

$\quad$ **if** $(p,q) \in \Omega_{\text{Asp}}$ **then**

$\quad\quad$ $\sigma(p,q) = -1$

$\quad$ **else**

$\quad\quad$ $\sigma(p,q) = 0$

$\quad$ **end if**

$\quad$ $\delta_k = \left( \sum_{i,j} \sigma(i,j) \right)^{-1}$ (Eq. 3)

$\quad$ $T_k = \sum_{l=1}^{k} \delta_l$

---



## 4   Sensitivity analysis

### 4.1   Parametric study

We performed a sensitivity analysis of the three asperity parameters ($\gamma_{\mathrm{asp}}$, $S_{\mathrm{a-Asp}}$, and $\pi_{\mathrm{asp}}$), in order to identify the best combination which produces the best approximation to real data, such as the maximum rupture area, $A_{\mathrm{syn}}$, and its related magnitude $M_{\mathrm{syn}}$. In order to investigate the influence of every single parameter, we statistically determined how the results vary with different parameter configurations.

#### 4.1.1   Percentage of transferred load, $\pi_{\mathbf{asp}}$

To explore the influence of $\pi_{\mathrm{asp}}$, we analyzed 12 values [$0.67 \leq \pi_{\mathrm{asp}} \leq 1.0$, with increments of 0.3]. The minimum $\pi_{\mathrm{asp}} = 0.67$ assigns the same value to an asperity cell, and to a background cell. On the other hand, $\pi_{\mathrm{asp}} = 1.0$ means that the load in a failed asperity cell is fully transmitted to their neighbors (ideal case no dissipative effects). Note that $\pi_{\mathrm{asp}} = 1$ does not represent real physical conditions since dissipative effects are ignored completely. On the other hand, if $\pi_{\mathrm{asp}} = 0.67$ (case 1) the asperity cells would transfer as much load as the cells in the background. The objective is to generate a load concentration within the asperity which corresponds to the largest magnitude. If the asperity cells transfer as much load as the background cells, no such load concentration can be obtained. As a result, we can expect that the mean $A_{\mathrm{syn}}$ for $\pi_{\mathrm{asp}} = 0.67$ (case 1) is the lowest value in comparison to all other cases.

The input data of this experiment is summarized in Table 2. We assigned a strength to the asperity ($R_{\mathrm{Asp}}$) $\gamma_{\mathrm{asp}} = 4 \pm 1$, and to the rest of the fault plane, we assigned a value of $\gamma = 1$. These values are chosen after experimental trials, which have shown that the difference is large enough to simulate a significant strength difference with low computational effort. To define the effective area, and the asperity size, we chose the values computed for the earthquake of 20/03/2012, Mw=7.4, in Rodríguez-Pérez and Ottemöller (2013): $A_{\mathrm{eff}} = 2944.2 \mathrm{km}^2$, and $S_{\mathrm{a}} = 0.26$. We defined the size of $\Omega$ consisting of in total $N_{\mathrm{cell}} = 10000$ cells. We carried out 50 simulations per $\pi_{\mathrm{asp}}$ configuration. In addition, we modified the random seed to have different initial load configurations, $\sigma_{(i,j)}$, to assure that the results over $\pi_{\mathrm{asp}}$ are independent of the initial load conditions $\sigma_{(i,j)}$.

#### 4.1.2   Strength parameter, $\gamma_{\mathbf{asp}}$

To perform the parametric study of $\gamma_{\mathrm{asp}}$, we configured two asperities embedded in $\Omega$. In this experiment, the total size is $\Omega = 200 \times 100$ cells. Afterwards, we located each asperity in the center of the two sub-domains $\Omega'$ of $100 \times 100$ cells. Fig. 5 shows a schematic representation of the domains $\Omega$, and $\Omega'$ used in this experiment.

The separation between both asperities remains constant. We chose a value of $\pi_{\mathrm{asp}} = 0.90$ to produce a large contrast between the asperity, and the rest of the fault plane ($\pi = 0.67$) (Monterrubio-Velasco et al., 2017). In order to analyze the influence of $\gamma_{\mathrm{asp}}$ (and $S_{\mathrm{a-Asp}}$), the asperity on the right side hand varying strength values, while the left asperity's strength remains constant. Finally, the maximum ruptured area, and magnitude generated in each $\Omega'$ is computed.





**Table 2.** Input data in order to carry out cases 1 to 12.

| Data | Value |
|---|---|
| Number of asperities | 1 |
| $\pi_{\mathrm{asp}}$ | 0.67 (case 1), 0.70 (case 2), 0.73 (case 3), |
| | 0.76 (case 4), 0.79 (case 5), 0.82 (case 6), |
| | 0.85 (case 7), 0.88 (case 8), 0.91 (case 9), |
| | 0.94 (case 10), 0.97 (case 11), and 1.0 (case 12). |
| Number of realizations | 50 |
| $N_{\mathrm{cell}}$ | 10000 |
| $\gamma_{\mathrm{asp}}$ | $4 \pm 1$ |
| $\pi$ | 0.67 |
| $\gamma$ | 1 |
| $\sigma_{\mathrm{th}}$ | 1 |
| $S_{\mathrm{a}}$ | 0.26 |
| $A_{\mathrm{eff}}$ | 2944.2 [km$^2$] |

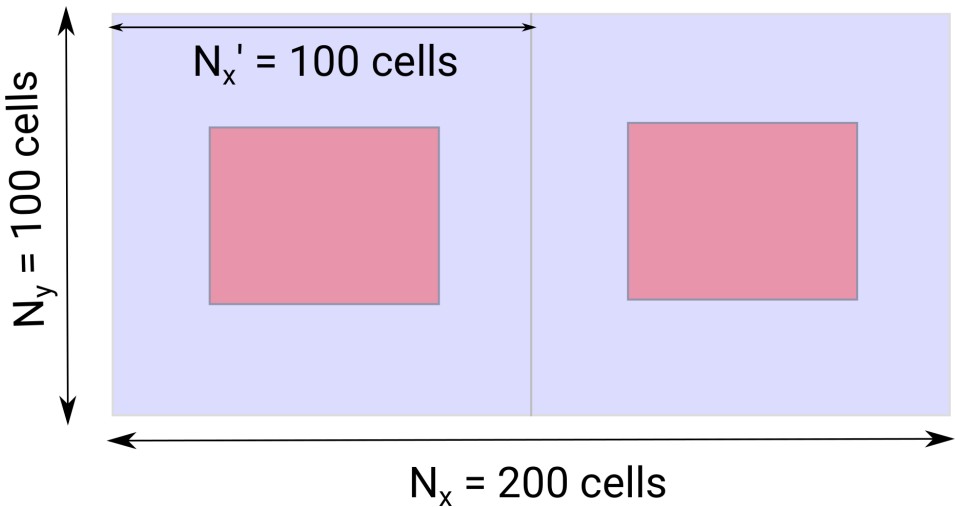

**Figure 5.** Schematic configuration for the parametric study of $\gamma_{\mathrm{asp}}$ and $S_{\mathrm{a-Asp}}$. The size of the domain $\Omega$ is $\Omega = 200 \times 100$ cells. Each asperity is located within the center of the two sub-domains $\Omega'$ of $100 \times 100$ cells. The strength parameter $\gamma_{\mathrm{asp}}$ and degree of heterogeneity for each asperity can be varied according to the material properties.

In order to explore how the system behaves when $\gamma_{\mathrm{asp}}$ changes, we analyzed 6 different values of $\gamma_{\mathrm{asp}} = [2 \pm 1, 5 \pm 1, 7 \pm 1, 9 \pm 1, 11 \pm 1, 14 \pm 1]$ (case 13 to 18). The input data used in this test is summarized in Table 3. We defined the same asperity





**Table 3.** Main input data in order to carry out cases 13 to 18.

| Data | Value |
|------|-------|
| Number of asperities | 2 |
| $\gamma_{\mathrm{asp}}$ | $2 \pm 1(case13), 5 \pm 1(case14), 7 \pm 1(case15),$ |
| | $9 \pm 1(case16), 11 \pm 1(case17), 14 \pm 1(case18)$ |
| $N_{\mathrm{cell}}$ | 20000 |
| $\pi_{\mathrm{asp}}$ | 0.90 |
| $\pi_{\mathrm{bkg}}$ | 0.67 |
| $\gamma_{\mathrm{bkg}}$ | 1 |
| $\sigma_{\mathrm{th}}$ | 1 |
| $S_{\mathrm{a1}}$ | 0.22 |
| $S_{\mathrm{a2}}$ | 0.22 |
| $A_{\mathrm{eff}}$ | 2944.2 [km$^2$] |

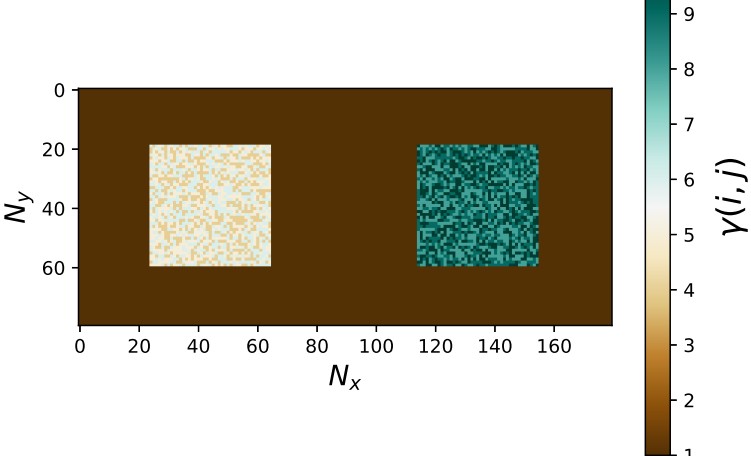

**Figure 6.** Example of the strength configuration $\gamma(i,j)$ for the sensitive analysis of $\gamma_{\mathrm{asp}}$. Two asperities with the same size $S_{\mathrm{a-Asp}} = 0.22$ are defined and embedded in $\Omega$ following the schema of Fig. 5. The conservation parameters are: $\pi_{\mathrm{asp}} = 0.90$ and $\pi_{\mathrm{asp}} = 0.67$. The color bar indicates different $\gamma(i,j)$ values. The left asperity (Asp. 1) contains constant properties, while the right asperity (Asp. 2) has variable strength values.

size for both, $S_{\mathrm{a1}} = S_{\mathrm{a2}} = 0.22$. In Fig. 6, we show an example of the spatial configuration of this analysis. The background strength is considered as constant $\gamma_{\mathrm{bkg}} = 1$, and the color bar indicates the $\gamma(i,j)$ values.





**Table 4.** Main input data in order to carry out cases 19 to 24.

| Data | Value |
| --- | --- |
| Number of asperities | 2 |
| $N_{\text{cell}}$ | 20000 |
| $\pi_{\text{asp}}$ | 0.90 |
| $\gamma_{\text{asp}}$ | $5 \pm 1$ |
| $\pi_{\text{bkg}}$ | 0.67 |
| $\gamma_{\text{bkg}}$ | 1 |
| $\sigma_{\text{th}}$ | 1 |
| $S_{\text{a2}}$ | 0.22 (case 19),0.28 (case 20),0.34 (case 21), |
| | 0.40(case 22), 0.46 (case 23), 0.52 (case 24) |

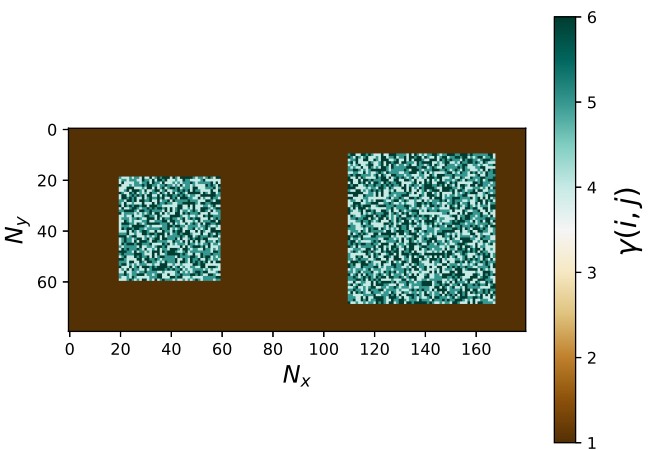

**Figure 7.** An example configuration of different asperity sizes, $S_{\text{a}-\text{Asp}}$. The color bar indicates the strength $\gamma(i,j)$ values used during the test.

### 4.1.3 Asperity size, $S_{\text{a}-\text{Asp}}$

The modification of the $S_{\text{a}-\text{Asp}}$ parameter was based on the same configuration as described in the previous section. We analyzed 6 different values of the asperity size $S_{\text{a}}$ (cases 19 to 24). In Fig. 7 we show an example of the asperity configuration where the left asperity has a constant size $S_{\text{a2}}$, while the size of the right one increases. In this experiment, we considered $\gamma_{\text{asp}} = 5 \pm 1$, and $\pi_{\text{asp}} = 0.90$. The main data related to these 6 cases is summarized in Table 4.



## 4.2 Model validation

We evaluated the capability of the model to reproduce the characteristics of 10 Mexican subduction earthquakes (8 shallow thrust subduction events, ST, and 2 intraslab subduction events, IN). The input data of the effective area $A_{\text{eff}}$, and the asperity ratio size, $S_{\text{a}}$, is given from waveform slip inversions and seismic source studies ($A_{\text{eff}} = L_{\text{eff}} \times W_{\text{eff}}$, and $S_{\text{a}} = A_{\text{a}}/A_{\text{eff}}$) shown in the database of the Mexican earthquake source parameters by Rodríguez-Pérez et al. (2018). This database includes results from two different methodologies: spectral analysis and finite-fault models. From the latter, the database provides estimations of effective fault dimensions, rupture velocity, source duration, number of asperities, stress and radiated seismic energy on the asperities and background areas. Slip solutions were obtained with teleseismic data for events with $6.4 < M_w < 8.2$.

The number of cells was $N_{\text{cell}} = 10000$ for a domain $\Omega$ of $100 \times 100$ cells. We modeled the size of $\Omega$ proportionally to the size of $L_{\text{eff}}$, and $W_{\text{eff}}$ for each scenario, according to the following equations, Eqs. 15 and 16:

$$N_{\text{x}} = \sqrt{\frac{N_{\text{cell}} L_{\text{eff}}}{W_{\text{eff}}}}, \tag{15}$$

$$N_{\text{y}} = \frac{W_{\text{eff}}}{L_{\text{eff}}} N_x, \tag{16}$$

where $N_x$ and $N_y$ are the number of cells in the x-axis, and in y-axis, respectively. As an example, Fig. 8 presents the size and aspect ratio of Ev. 3 and Ev. 5 (Table 5).

In some cases, the number of asperities computed in Rodríguez-Pérez et al. (2018) are greater than 1. However, as a first approximation we simplified the problem by modeling only one asperity per earthquake.

In order to study how the asperity size $S_{\text{a}}$ affects the maximum ruptured area, we randomly modified the size as

$$S_{\text{a}-\text{Asp}} = S_{\text{a}} + (\alpha * (S_{\text{a}}/2)), \tag{17}$$

where $0 < \alpha < 1$ is a random value. We introduce this assumption because we want to avoid a preconceived final size. In future trials it may be useful to consider the inner uncertainties of finite-fault models. The asperity aspect ratio follows the same proportion as the effective area, $\frac{\Omega_x}{\Omega_y} = \frac{N_{\text{x(Sa)}}}{N_{\text{y(Sa)}}}$ (Fig. 8).

We carried out 50 realizations per event (Table 5) changing the size $S_{\text{a}-\text{Asp}}$ in each one (Eq. 17).

### 4.2.1 Modelling the rupture area and magnitude of 10 subduction earthquakes

In this case the number of cells is $N_{\text{cell}} = 10000$ cells ($100 \times 100$). We carried out 50 executions per event and in each execution we randomly changed the size $S_{\text{a}-\text{Asp}}$ following Eqs. 15, 16, and 17. The input data of the 10 modeled earthquakes of Table 5 are summarized in Table 6.



**Table 5.** The finite-fault source parameters used in this work. $W_{\mathrm{eff}}$ and $L_{\mathrm{eff}}$ are the effective fault dimensions (width and length, respectively, according to Mai and Beroza (2000)). $A_{\mathrm{real}}$ is the asperity area, $A_{\mathrm{eff}}$ is the effective rupture area ($W_{\mathrm{eff}} \times L_{\mathrm{eff}}$). $Duration$ is the rupture duration computed from the slip inversion, $N_{\mathrm{a}}$ is the number of asperities and $V_r$ is the rupture velocity. $Ratio$ is the aspect ratio of the fault area. The type of the event is labeled $ST$ for shallow thrust and $IN$ for intraslab events.

| $Ev.ID$ | Date | $M_w$ | $L_{\mathrm{eff}}$[km] | $W_{\mathrm{eff}}$[km] | $Ratio$ | $S_a = A_{\mathrm{real}}/A_{\mathrm{eff}}$ | $Duration$[s] | $V_r$[km/S] | Type | $N_a$ | $Reference$ |
|---|---|---|---|---|---|---|---|---|---|---|---|
| 1 | 07/06/1982 | 7.0 | 34.47 | 17.81 | 1.94 | 0.23 | – | 3.2 | ST | 1 | (Rodríguez-Pérez and Zúñiga, 2016) |
| 2 | 19/09/1985 | 8.1 | 158.62 | 115.04 | 1.38 | 0.31 | – | 2.6 | ST | 2 | (Mendoza, 1989) |
| 3 | 30/04/1986 | 6.8 | 38.31 | 37.16 | 1.03 | 0.26 | 22 | 2.5 | ST | 1 | (Rodríguez-Pérez and Ottemöller, 2013) |
| 4 | 14/09/1995 | 7.4 | 68.80 | 46.61 | 1.48 | 0.23 | 32 | 2.5 | ST | 1 | (Rodríguez-Pérez and Ottemöller, 2013) |
| 5 | 09/10/1995 | 8.0 | 169.65 | 59.25 | 2.86 | 0.27 | 92 | 2.8 | ST | 2 | (Rodríguez-Pérez and Ottemöller, 2013) |
| 6 | 18/04/2002 | 6.7 | 23 | 13.88 | 1.66 | 0.24 | 30 | 2.2 | ST | 2 | (Rodríguez-Pérez and Ottemöller, 2013) |
| 7 | 20/03/2012 | 7.4 | 54.94 | 53.59 | 1.03 | 0.26 | 30 | 2.7 | ST | 1 | (Rodríguez-Pérez and Ottemöller, 2013) |
| 7a | | 7.4 | 51.42 | 55.47 | 0.93 | 0.21 | – | 1.8 | ST | 1 | USGS |
| 7b | | 7.4 | 40.03 | 44.60 | 0.89 | 0.21 | – | 2.0 | ST | 1 | (Wei, 2012) |
| 8 | 11/04/2012 | 6.5 | 21.95 | 21.84 | 1.04 | 0.23 | 15 | 2.8 | ST | 1 | (Rodríguez-Pérez and Ottemöller, 2013) |
| 9 | 08/09/2017 | 8.2 | 125.95 | 71.13 | 1.77 | 0.34 | – | 2.0 | IN | 3 | USGS |
| 10 | 19/09/2017 | 7.1 | 34.47 | 36.12 | 0.95 | 0.32 | – | 2.2 | IN | 1 | USGS height |

**Table 6.** Main data used for Ev.1 to Ev. 10.

| Data | Value |
|---|---|
| Number of asperities | 1 |
| $N_{\mathrm{cell}}$ | 10000 |
| $\pi_{\mathrm{asp}}$ | 0.90 |
| $\gamma_{\mathrm{asp}}$ | $5 \pm 1$ |
| $\pi_{\mathrm{bkg}}$ | 0.67 |
| $\gamma_{\mathrm{bkg}}$ | 1 |
| $\sigma_{\mathrm{th}}$ | 1 |
| $S_{\mathrm{a}}$ | see Table 5 |
| $S_{\mathrm{a-Asp}}$ | Eqs. 17 |
| $A_{\mathrm{eff}}$ | see Table 5 |





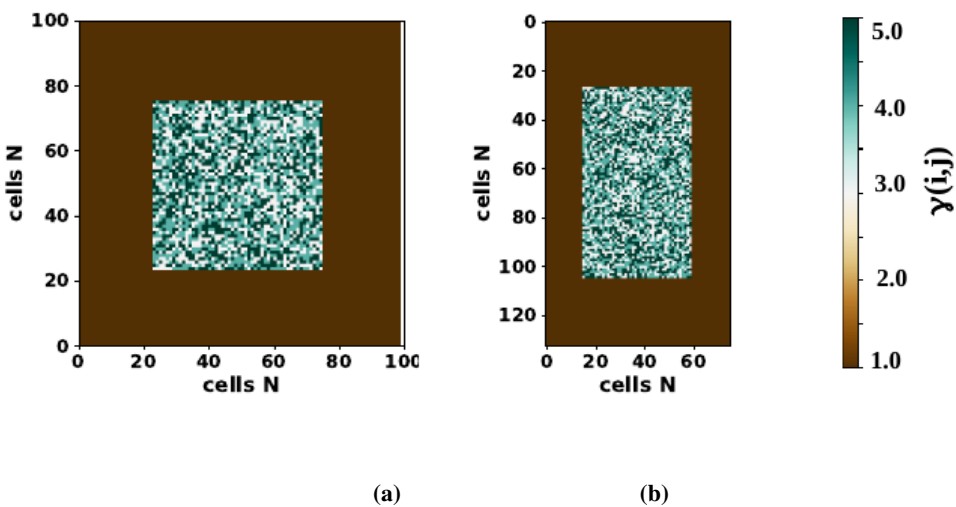

(a)                                                    (b)

**Figure 8.** Example of the domain configuration $\Omega$, considering $L_{\mathrm{eff}}$ and $W_{\mathrm{eff}}$. (a) Example configuration of Event 3 and (b) Example configuration of Event 5. The required data can be found in table 5.

### 4.2.2    Case-study (Oaxaca, $M_w = 7.4$, 20/03/2012): Using different effective areas $A_{\mathrm{eff}}$ for the same event

As reported in Rodríguez-Pérez et al. (2018) for some events, there are several solutions which allows us to analyze the variability in the estimated source parameters (see parameters of events 7, 7a, 7b in Table 5). In this study, we applied TREMOL to study how the ruptured area and the assessed magnitude changes when we use different input data to model the same

earthquake. The data related to these three events is summarized in Table 7.

### 4.2.3    Assessing a future earthquake in the Guerrero seismic gap: rupture area and magnitude

We apply our method for the estimation of possible future earthquakes. In particular, to compute the expected magnitude, since TREMOL may offer new insights for future hazard assessments. We carried out a statistical test to assess the size of an earthquake that may occur in the Guerrero seismic gap (GG) region.

As input parameters, we used the area found by Singh and Mortera: $L_{\mathrm{eff}} = 230\mathrm{km} \times W_{\mathrm{eff}} = 80$ km. We defined the asperity size ratio $S_{\mathrm{a}}$ as proposed by Somerville et al. (2002) for regular subduction zone events (SB), based on average slip, $S_{\mathrm{a}} = 0.25$. Singh and Mortera, Astiz et al. (1987), and Astiz and Kanamori (1984), proposed a probable maximum magnitude for this region of $M_w \approx 8.1 - 8.4$. Therefore, using the effective rupture area ($L_{\mathrm{eff}}$, $W_{\mathrm{eff}}$ and $S_{\mathrm{a}}$), we executed the algorithm as in previous sections. The input data related to this analysis is summarized in Table 8. Likewise, we want to estimated the duration

$D_{\mathrm{aval}}$ of the event. To compute this value, we used a mean of the $V_{\mathrm{r}}$ from Table 5.



**Table 7.** Main data for the case-study Ev. 7 test, Ev. 7a test, and Ev. 7b test.

| Data | Value |
|------|-------|
| Number of asperities | 1 |
| $N_{\text{cell}}$ | 10000 [cells] |
| $\pi_{\text{asp}}$ | 0.90 |
| $\gamma_{\text{asp}}$ | $5 \pm 1$ |
| $\pi_{\text{bkg}}$ | 0.67 |
| $\gamma_{\text{bkg}}$ | 1 |
| $\sigma_{\text{th}}$ | 1 |
| $S_{\text{a-Asp}}$ | (Eqs. 17) |
| $S_{\text{a}}$ | see Table 5 (Ev. 7, 7a, and 7b) |
| $A_{\text{eff}}$ | see Table 5 (Ev. 7, 7a, and 7b) |

**Table 8.** Main data for assessing a future earthquake in the Guerrero seismic gap (GG event).

| Data | Value |
|------|-------|
| Number of asperities | 1 |
| $N_{\text{cell}}$ | 10000 [cells] |
| $\pi_{\text{asp}}$ | 0.90 |
| $\gamma_{\text{asp}}$ | $5 \pm 1$ |
| $\pi_{\text{bkg}}$ | 0.67 |
| $\gamma_{\text{bkg}}$ | 1 |
| $\sigma_{\text{th}}$ | 1 |
| $S_{\text{a-Asp}}$ | (Eqs. 17) |
| $S_{\text{a}}$ | 0.25 |
| $A_{\text{eff}}$ | 18400 [km$^2$]) |

# 5 Results

## 5.1 Parametric study

### 5.1.1 Percentage of transferred load, $\pi_{\text{asp}}$

Fig. 9 shows the mean (black dots) of the maximum ruptured area $A_{\text{syn}}$ including the upper and lower limits of the standard deviations (blue squares) after the execution of all 12 cases (Table 2) with 50 realizations. The value of $A_{\text{syn}}$ is related to the largest produced cluster in $\Omega$. There are two dominant tendencies identifiable:

1. If $\pi_{\text{asp}} < 0.76$, the mean of the maximum ruptured area increases continuously more than one order of magnitude – from 15 to $\approx 500\ \text{km}^2$, i.e. an increase of 3333 %. The standard deviation of $A_{\text{syn}}$ for $\pi_{\text{asp}} = 0.7$ is $\approx 35\ \text{km}^2$ (100 % error).



2. If $\pi_{\mathrm{asp}} \geq 0.76$ (cases 4 to 12), the $A_{\mathrm{syn}}$ values remain essentially constant ($\approx 500\ \mathrm{km}^2$). Likewise, the upper and lower limit vary around the same order. The standard deviation for this interval is $\approx 100\ \mathrm{km}^2$ (20% error).

Using the mean of $A_{\mathrm{syn}}$ obtained in each case, we computed the corresponding magnitude. The results are given as mean and the standard deviation of the maximum magnitude in Fig. 10 for all twelve cases (see Table 2). Due to the fact that ruptured area and magnitude are correlated (see Eqs. 7, 8 and 9), the pattern in Fig. 10 is very similar to the one in Fig. 9.

Overall, there are three aspects observable:

1. If $\pi_{\mathrm{asp}} \geq 0.76$ (cases 4 to 12), the mean magnitudes show a steady value ($\approx 7.2$).

2. If $0.70 \leq \pi_{\mathrm{asp}} < 0.76$ a transition with an increasing trend with the largest standard deviation is visible.

3. If $\pi_{\mathrm{asp}} = 0.67$ (case 1), the mean of the maximum magnitude is the lowest.

In this experiment, the initial value of $S_{\mathrm{a}} = 0.26$ remains constant, *i.e.* the asperity size does not increase randomly (red line in Fig. 11). After executing all configurations, we computed the ratio of $S_{\mathrm{a-Asp}} = A_{\mathrm{syn}}/A_{\mathrm{eff}}$, relating to the largest ruptured area. We show the mean, and standard deviation of this ratio $S_{\mathrm{a-Asp}}$ in Fig. 11. We observed that the ratio of $S_{\mathrm{a-Asp}}$ is always $\approx 0.10$ lower than $S_{\mathrm{a}}$.

### 5.1.2 Strength parameter, $\gamma_{\mathrm{asp}}$

For each value of $\gamma_{\mathrm{asp}}$ (Table 3), we performed 50 executions while changing the initial strength parameter of the asperity $\gamma_{\mathrm{asp}}$ (Fig. 3b). Likewise, we computed the maximum magnitude obtained for each $\Omega'$. Fig. 12 indicates the mean and standard deviation of the computed maximum magnitude in dependence on $\gamma_{\mathrm{asp}}$. The upper subplot (blue markers) shows the results for the left (constant) asperity (Asp. 1). The lower subplot (red markers) shows the results for the right (variable) asperity (Asp. 2).

We observe in Fig. 12 that the mean magnitude remains essentially independent for $\gamma_{\mathrm{asp}} > 5 \pm 1$. Additionally, the error bars slightly decrease while $\gamma_{\mathrm{asp}}$ increases. Another observation is that when $\gamma_{\mathrm{asp}} = 2 \pm 1$ the average of the maximum magnitude is the lowest in both asperities. Moreover, there is a transition zone for $2 \pm 1 \leq \gamma_{\mathrm{asp}} \leq 5 \pm 1$. We observed that $\gamma_{\mathrm{asp}} > 5 \pm 1$ has a limited influence on the results of the maximum magnitude. The maximum magnitude of $\gamma_{\mathrm{asp}} = 14 \pm 1$ is approx. 0.3 magnitudes larger than the one of $\gamma_{\mathrm{asp}} = 5 \pm 1$.

### 5.1.3 Asperity size, $S_{\mathrm{a-Asp}}$

Fig. 13 shows the mean magnitude and standard deviation as a function of asperity size. The first asperity with the fixed size indicates a relative constant magnitude of approx. 7.4. Conversely, the second asperity with variable size produces only a slight increase in magnitude. The magnitude of $S_{\mathrm{a-Asp}} = 0.52$ is approx. 0.5 magnitudes larger than the one of $S_{\mathrm{a-Asp}} = 0.22$.





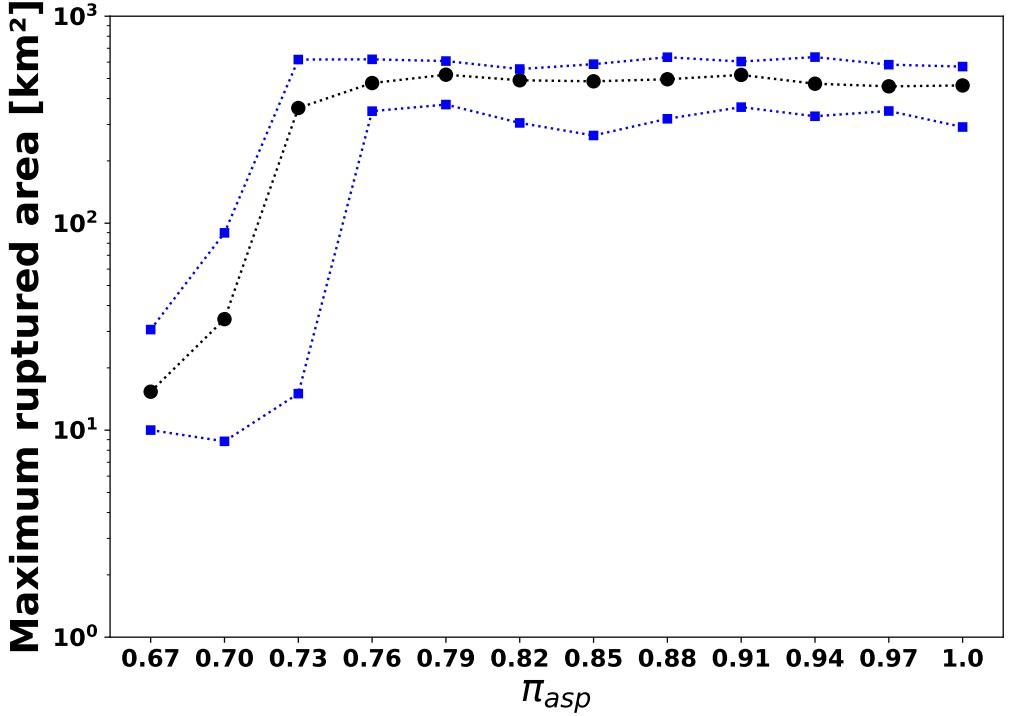

**Figure 9.** Mean of the maximum rupture area $[\mathrm{km}^2]$, $A_{\mathrm{syn}}$ for different values of $\pi_{\mathrm{asp}}$ depicted as black circles. The minimum and maximum limits of the rupture area are represented by blue squares.

## 5.2 Model validation

### 5.2.1 Modelling 10 Mexican subduction zone earthquakes

Based on the observations described in the previous section, we used $\gamma_{\mathrm{asp}} = 5 \pm 1$, and $\pi_{\mathrm{asp}} = 0.90$ in order to validate the model. We chose $\gamma_{\mathrm{asp}} = 5 \pm 1$ because it represents the strength interval of $5 \pm 1 \leq \gamma_{\mathrm{asp}} \leq 14 \pm 1$ with less computational costs.

5  We chose $\pi_{\mathrm{asp}} = 0.90$ because it represents the relatively constant magnitude for the parameter range $0.76 \leq \pi_{\mathrm{asp}} \leq 0.90$. In addition, $\pi_{\mathrm{asp}} = 0.90$ enables to obtain the best approximation to the ratios of $S_{\mathrm{a-Asp}}$. Both parameter choices ensure an appropriate reproduction of the asperity rupture area, the maximum magnitude and least computational payload.

Fig. 14 depicts a comparison between the (real) asperity area $A_{\mathrm{real}}$ (Table 5), and the area of the largest simulated earthquake, $A_{\mathrm{syn}}$. We plot the mean (blue dots), the minimum (green triangles), and the maximum (red triangles) of all 50 realizations for

10  each real earthquake event. Black squares represent the real asperity size. The results in Fig. 14 point out that $A_{\mathrm{syn}}$ is almost identical to $A_{\mathrm{real}}$ from Table 5 for the majority of earthquakes. Only three events show significant differences between synthetic and realistic maximum rupture area. Even in these cases, however, $A_{\mathrm{real}}$ is located within the upper and lower limit of $A_{\mathrm{syn}}$.





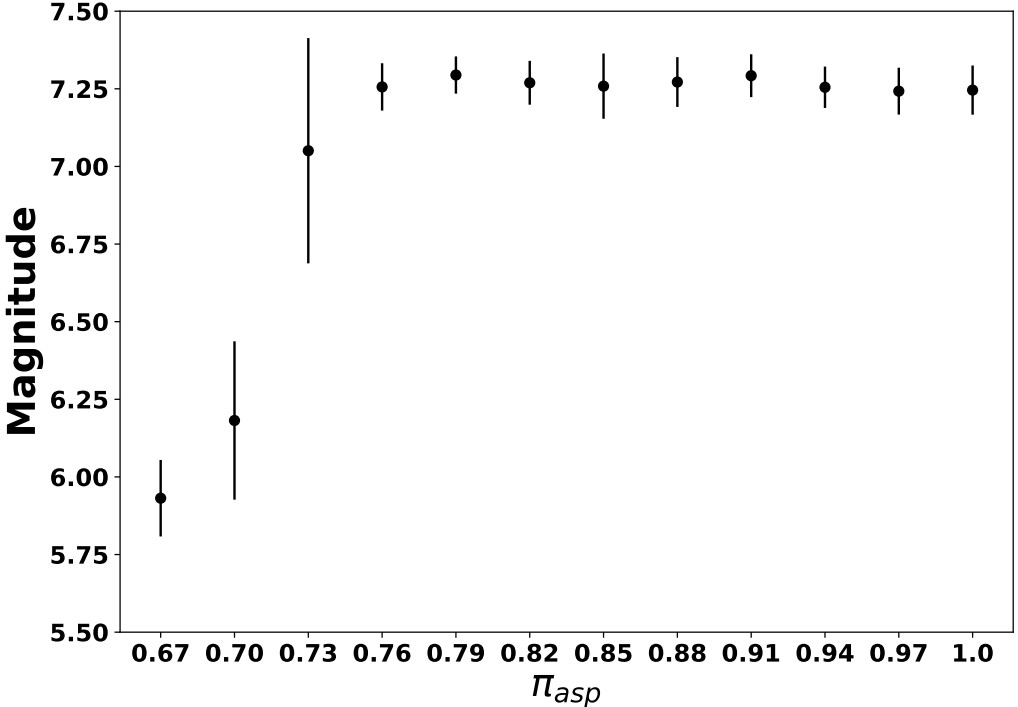

**Figure 10.** Mean and standard deviation of the maximum magnitude over 50 realizations depending on $\pi_{\mathrm{asp}}$.

Fig. 15 shows the statistical results of the synthetic maximum magnitude, $M_{\mathrm{syn}}$, determined for all 10 events. The real magnitudes from Table 5 are given as red markers. Black circles indicate the mean of $M_{\mathrm{syn}}$ of 50 realizations using Eq. 7, whereas blue and green markers indicate the magnitude following the equations 8 and 9, respectively. The error bars represent the standard deviation. We observed that the statistical parameters computed with TREMOL fit the magnitudes shown in Table

5. 5. However, the computed magnitudes depend on the scale relation employed (Eq. 7, Eq. 8, and Eq. 9). Fig. 16 includes the mean of the three scale relations. Overall, the mean magnitude $\overline{M}_{\mathrm{syn}}$ and the expected magnitude $M_w$ show similar values. Given that the difference between the mean and the expected value (Table 5) is lower than $\Delta M_w < 0.5$ for the 10 events, we can affirm that the results of assessing the magnitude by means of TREMOL using a randomly modified asperity size, $S_{\mathrm{a-Asp}}$ (Eq. 17), are reasonable.

10. Fig. 17 shows the real ratio size $S_{\mathrm{a}}$ from table 5 (black squares) in comparison to the mean of the largest simulated earthquake, $S_{\mathrm{syn}}$ (blue squares). The standard deviation is represented as error bars. The results indicate that in most of the cases the computed $S_{\mathrm{syn}}$ range fits the expected $S_{\mathrm{a}}$ well. Note that for the Events 3, 7, and 8 the mean values are lower than the reported $S_{\mathrm{a}}$, while $S_{\mathrm{syn}}$ is overestimated for Events 2, 5 and 9. For Events 1, 4, 6 and 10 the estimated value of $S_{\mathrm{syn}}$ coincides with the expected one. However, the error bars enclose the expected values in all cases (Fig. 17). Moreover, if we compare Fig. 17



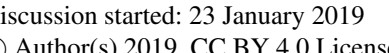

**Figure 11.** Mean and standard deviation of the ratio $S_{\mathrm{syn}} = A_{\mathrm{syn}}/A_{\mathrm{eff}}$ over 50 realizations for different values of $\pi_{\mathrm{asp}}$. The red line indicates the asperity ratio $S_{\mathrm{a}}$ computed for event 7 (Table 5).

with Fig. 11 we observe that the employed strategy of randomly increasing asperity size (using Eq. 17) generate rupture areas similar to the ones proposed by Rodríguez-Pérez et al. (2018).

We also computed an equivalent rupture duration, $D_{\mathrm{Aval}}$ using the equation proposed by Geller (1976) to calculate the rise time (Eq. 13 and 14). Rodríguez-Pérez and Ottemöller (2013) determined the rupture velocity $V_{\mathrm{r}}$ (Ev. 3-8), which is a useful

parameter in order to validate our results. Fig. 18 shows the results of this analysis. In red we plot the values $V_{\mathrm{r}}$ calculated by Rodríguez-Pérez and Ottemöller (2013), and in blue the $D_{\mathrm{Aval}}$ based on Eq. 13 with $V_{\mathrm{r}}$ provided by Table 5. The equivalent $D_{\mathrm{Aval}}$ using the Eqs. 13 and 14 is printed in black. In cases where we have the reference values, $V_{\mathrm{r}}$, computed by Rodríguez-Pérez and Ottemöller (2013), we observe that the reference value are always larger than the modelled $D_{\mathrm{Aval}}$ values. However, is worth to note that $V_{\mathrm{r}}$ is the mean rupture time that considers the rupture of the whole effective area ($A_{\mathrm{eff}}$). In the simulated

events, $D_{\mathrm{Aval}}$, we only consider the rupture length of the largest rupture cluster $A_{\mathrm{syn}}$. So, smaller values than those proposed in Rodríguez-Pérez and Ottemöller (2013) are expected. Nevertheless, the rupture duration shows a clear dependency on the magnitude.

### 5.2.2  Case-study (Oaxaca, $M_w = 7.4$, 20/03/2012)

In the cases where several effective rupture areas were proposed by different studies (see Table 5), it is possible to assess which

set of parameters is better in order to simulate an event by means of TREMOL. We tested TREMOL by using three different combinations of $L_{\mathrm{eff}}$, $W_{\mathrm{eff}}$ and $S_{\mathrm{a}}$ according to results for Ev. 7 in Table 5. A comparison of these three combinations is visualized in Fig. 19: (a) shows the comparison of the ruptured areas, $A_{\mathrm{real}}$ and $A_{\mathrm{syn}}$; (b) shows the mean and standard deviation of the maximum magnitude, $M_{\mathrm{syn}}$, in comparison to the reference magnitude; (c) shows the ratio $S_{\mathrm{syn}}$ of the simulated events



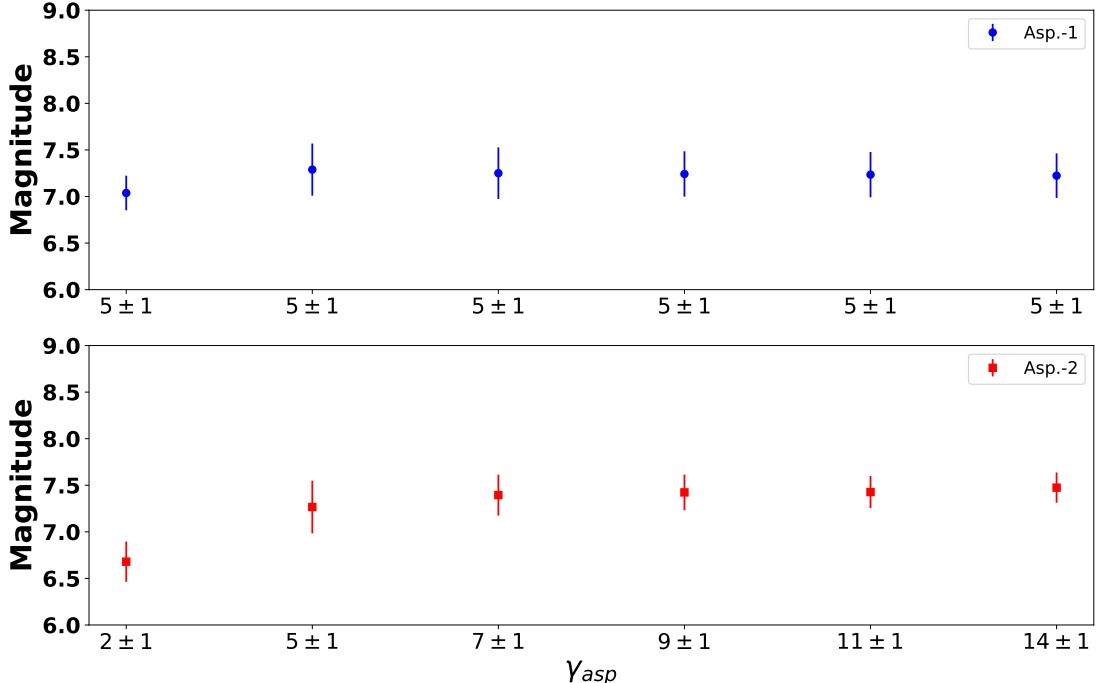

**Figure 12.** Statistical results of $\gamma_{\mathrm{asp}}$ for a configuration similar to figure 6. Markers represent the mean value while the error bars indicate the standard deviation for all 50 executions considering different initial strength configurations. The red markers correspond to the results of the left asperity (Asp. 1) and blue markers to the results of the right asperity (Asp. 2) (Fig. 6). The strength of the left asperity is kept constant, whereas the strength of the right asperity is variable.

compared to $S_{\mathrm{a}}$ the real scenarios. Although, the three combinations express similar results, the closest approximation between real and synthetic data is generated based on the data by Rodríguez-Pérez and Ottemöller (2013) (Ev. 7).

### 5.2.3 Assessing a future earthquake in the Guerrero seismic gap: rupture area and magnitude

In Fig. 20 (a), we compare the mean of maximum ruptured area, $A_{\mathrm{syn}}$, including error bars with the reference area, $A_{\mathrm{a}}$. The

5 rupture area computed in TREMOL shows a possible range from $4000\,\mathrm{km}^2$ to $7000\,\mathrm{km}^2$. This interval is based on a considered size of $S_a = 0.25$. In the subplot of Figure 20 (b), we estimated the duration $D_{\mathrm{aval}}$ of the rupture event. The results in Fig. 20 (b) indicate that the duration $D_{\mathrm{aval}}$ is similar to that of the other events of magnitude $M_w \approx 8$. The duration may range from 80 to 110 seconds, while a rupture duration between 90 and 100 seconds is most likely. Fig. 20 (c) shows the mean of the estimated magnitude using Eqs. 7, 8, and 9. TREMOL ejects a possible range of $8.1 \leq M_w \leq 8.5$, which matches the proposed

10 value by Singh and Mortera; Astiz et al. (1987); Astiz and Kanamori (1984) of $M_w \approx 8.1 - 8.4$.

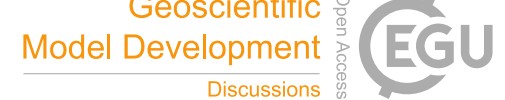



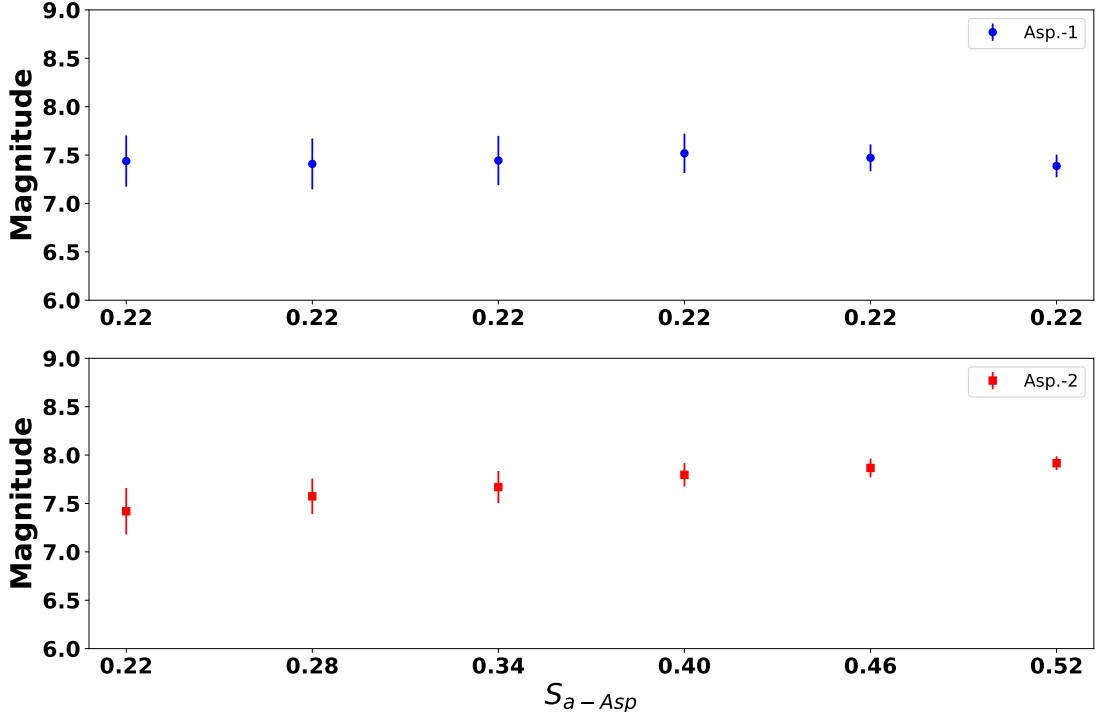

**Figure 13.** Statistical results of $S_{\mathrm{a-Asp}}$ for a configuration similar to figure 7. The markers indicate mean and standard deviation for 50 realizations. Red markers correspond to the results of the right and variable asperity and blue markers to the results of the right and stable asperity.

# 6 Discussion

## 6.1 Parametric study

### 6.1.1 Percentage of transferred load, $\pi_{\mathrm{asp}}$

In the results, there were two dominant tendencies visible: (1) $\pi_{\mathrm{asp}} < 0.76$, and (2) $0.76 \leq \pi_{\mathrm{asp}}$. If $\pi_{\mathrm{asp}} < 0.76$ the mean of
5  the maximum ruptured area increased continuously more than one order of magnitude from 15 to $\approx 500 \mathrm{km}^2$, i.e. an increase of 3333 %. Therefore, the range of $\pi_{\mathrm{asp}}$ is both crucial and sensitive. A parameter increase of only 15 %, affects the size of the biggest earthquake within the system by 3333 %. Considering the large standard deviation of $\approx 35 \mathrm{~km}^2$ (100 % error) a parameter configuration based on $\pi_{\mathrm{asp}} < 0.76$ would be unsuitable for further simulations due to the unstable properties obtained for that range. The second tendency, however, offers the possibility to determine a stable conservation parameter
10  which can be freely chosen in the range of $0.76 \leq \pi_{\mathrm{asp}} \leq 1.0$. The stable state of maximum rupture area is caused by a self-organized critical Avalanche size of $A_{\mathrm{crit}} \approx 500 \mathrm{km}^2$ based on a grid of $100 \times 100$ cells with $A_{\mathrm{eff}} = 2944.2 \mathrm{km}^2$, and $S_{\mathrm{a}} = 0.26$.



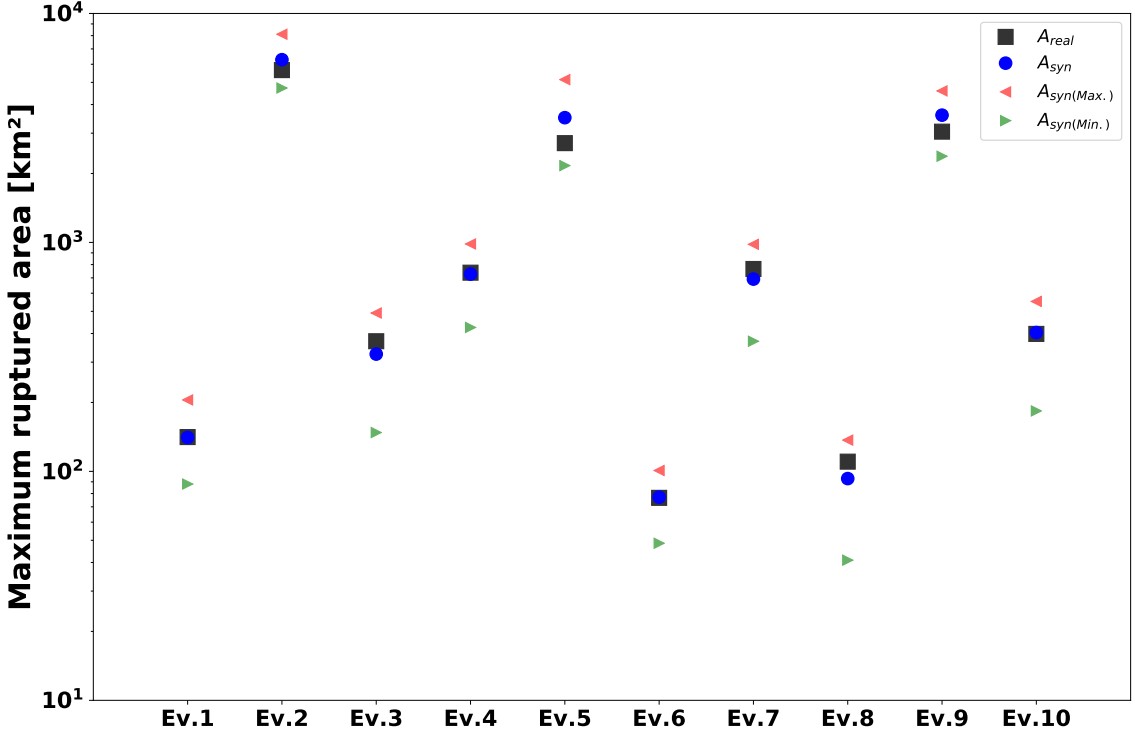

**Figure 14.** Comparison between the real asperity area $A_{\mathrm{real}}$ and the synthetic values (mean and standard deviation) of the largest ruptured event $A_{\mathrm{syn}}$. Black squares depict the real asperity area from table 5, whereas blue circles indicate the mean area of 50 executions. Red and green triangles represent the maximum and minimum $A_{\mathrm{syn}}$.

As soon as $A_{\mathrm{crit}}$ is achieved by the system, the largest Avalanche will stop to increase in size, whereas other Avalanches within the system will be favoured to grow. On the other hand, this means that TREMOL breaks the asperity rather in patches than completely during one unique rupture event (see Figs. 4 and 11). This last condition is reasonable considering that the algorithm of FBM used in TREMOL favors clustering the rupture of cells. Therefore, it is reasonable that some cells remain outside of a unique rupture group because they do not satisfy the failure conditions. As a consequence, we think that it is necessary to define an initial area greater than the expected area of the asperity, where the asperity rupture can occur. This result also justifies the proposed Eq. 17, where the size of the asperity increases randomly up to 50 % larger than the value proposed by Rodríguez-Pérez et al. (2018). Future studies may be useful to better determine the influence of $A_{\mathrm{crit}}$.





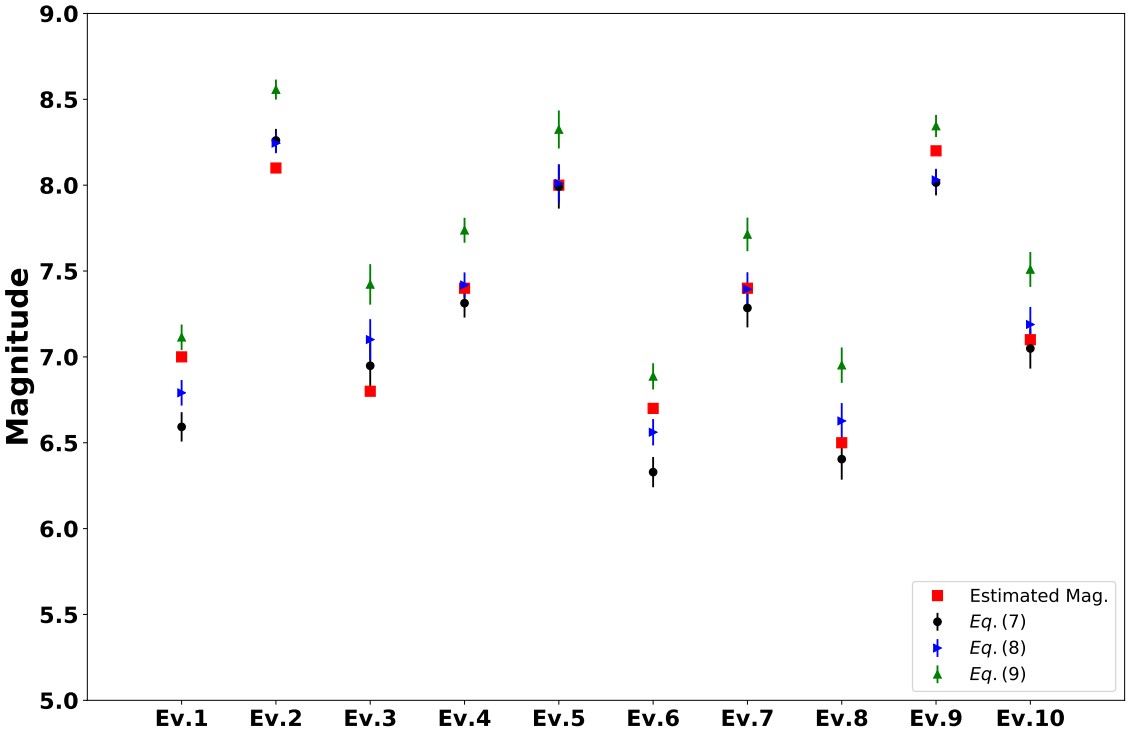

**Figure 15.** Statistical results of the maximum magnitude for the events from table 5. Red squares depict the real estimated magnitudes from table 5, while black circles, blue triangles and green triangles represent the synthetic mean magnitude of 50 executions following Eqs. 7, 8 and 9, respectively. The error bars are the standard deviation of the scale relations.

The parametric study indicates that the largest rupture $\pi_{\mathrm{asp}}$ is produced as long as it is within the range $0.76 < \pi_{\mathrm{asp}} < 1.0$. So even though as $\pi_{\mathrm{asp}}$ increases, large rupture clusters are generated because a large amount of load is transferred to the neighboring cells, producing critical local load concentrations in the system, and the particular lower bound is critical.

### 6.1.2   Strength parameter, $\gamma_{\mathrm{asp}}$

5   For the strength parameter $\gamma_{\mathrm{asp}}$ there were two tendencies visible as well:

1. An unstable transition zone of $2 \pm 1 \leq \gamma_{\mathrm{asp}} \leq 5 \pm 1$ where the maximum rupture has a strong variation. Therefore, a strength value within this range should be avoided.



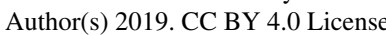


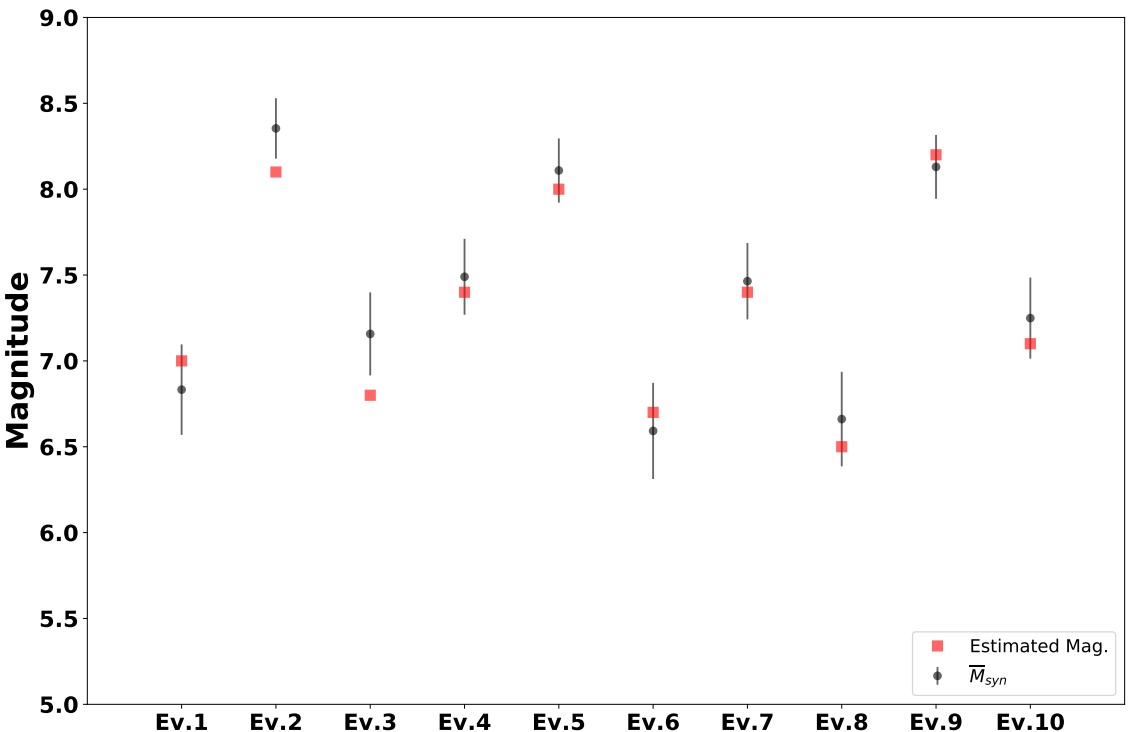

**Figure 16.** Statistical results of the maximum magnitude for the events from table 5. Red squares represent the magnitudes from table 5, whereas black circles, the mean magnitude ($\overline{M}_{\mathrm{syn}}$) value of all 50 executions following Eqs. 7, 8, and 9. The error bars stand for the standard deviation of the mean for the three scale relations.

2. A stable zone of $5 \pm 1 < \gamma_{\mathrm{asp}} < 14 \pm 1$, where $\gamma_{\mathrm{asp}}$ can be freely chosen. However, due to computational costs it is recommended to use the lowest value of $\gamma_{\mathrm{asp}} = 5 \pm 1$, since the number of necessary time-steps to activate the whole asperity increases strongly with the applied asperity strength (see Algorithm 1).

### 6.1.3   Asperity size, $S_{\mathrm{a-Asp}}$

5    The results of Fig. 13 indicate that asperity size has a significant influence on the maximum magnitude. We emphasize the importance of these results because they show that the parameter $S_{\mathrm{a-Asp}}$ is critical to control the generated magnitude. At the same time, these results provide the appropriate range of values that TREMOL requires to do a reasonable assessment of the maximum rupture area and magnitude of an earthquake.



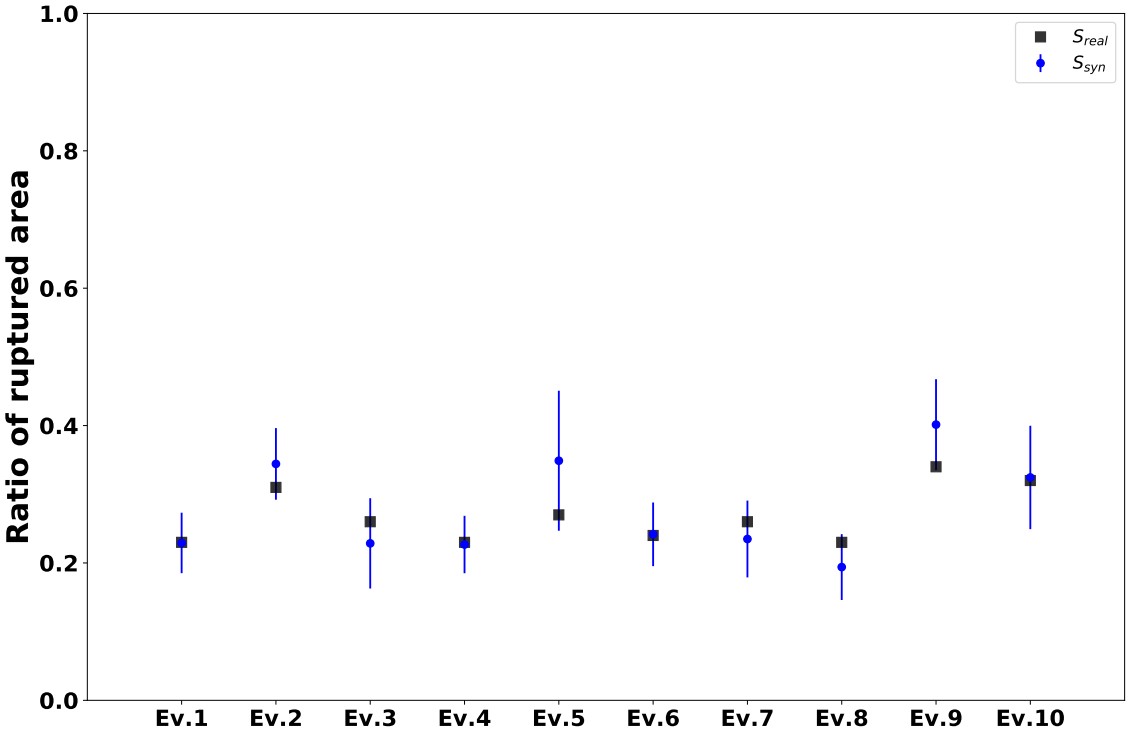

**Figure 17.** Proportion of simulated ruptured area occupied by the largest *Avalanche*, $S_{\mathrm{syn}}$, in comparison with the real ratio size $S_{\mathrm{a}}$ from table 5. The real ratio size $S_{\mathrm{a}}$ from table 5 is represented by black squares and the mean of the largest simulated earthquake, $S_{\mathrm{syn}}$ by blue squares. The standard deviation is represented as error bars.

## 6.2 Model validation

The model validation by means of 10 different subduction earthquakes showed that TREMOL is capable of reproducing rupture area and magnitude appropriately – by means of only few input data – in comparison to the results from inversion studies. The computed rupture duration by TREMOL differs somehow from the reference values. The reason may be that the calculation of the rupture duration is based on the largest (critical) rupture area which is not equal to the available asperity area (see Fig. 11 and 6.1.1). Nevertheless, the rupture duration shows a clear dependency on the magnitude.

Since TREMOL only requires few input data, it is a powerful tool to simulate future earthquakes, such as those which might take place in the Guerrero Gap region. The determination of the magnitude of an earthquake based on the asperity area, depends on the used scale relation. Nevertheless, the rupture area is not model-sensitive.





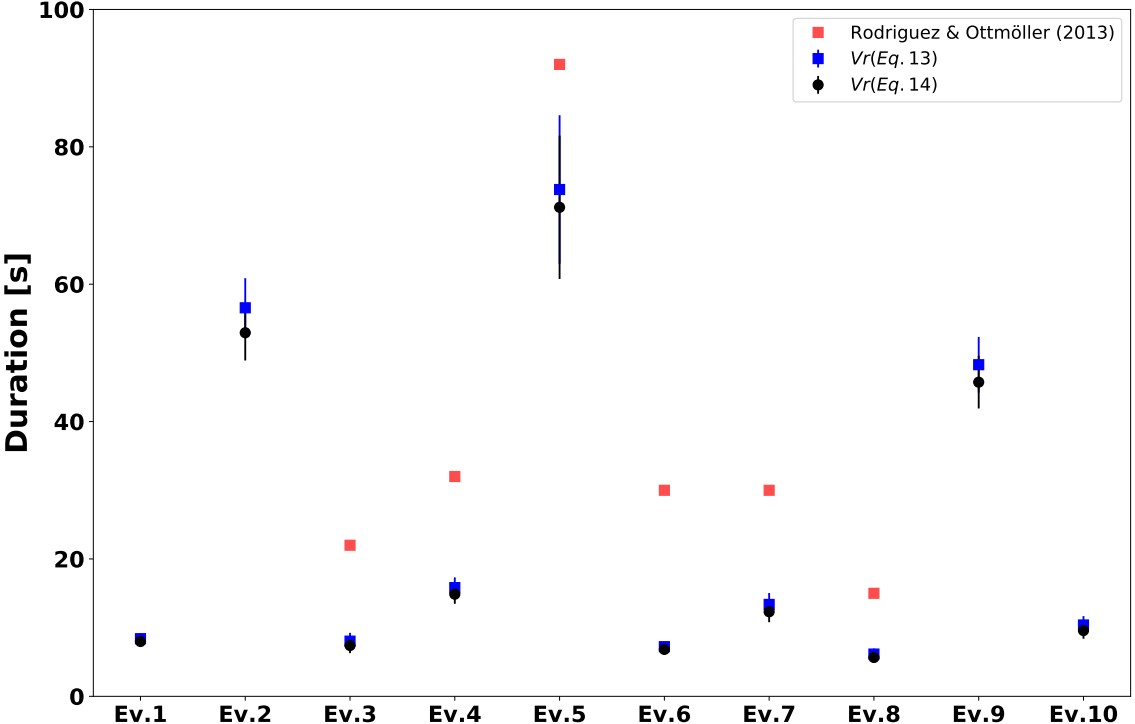

**Figure 18.** Equivalent rupture duration $D_{\mathrm{Aval}}$ [seconds] calculated via the rupture velocity by using the size of the largest rupture cluster. Red squares represent the reference values proposed by Rodríguez-Pérez and Ottemöller (2013), while blue squares and black circles depict the synthetic rupture duration computed by means of $V_r$ based on Eq. 13 and Eq. 14, respectively.

### 6.3 TREMOL: Advantages and disadvantages

The algorithm of TREMOL enables the model to store stress history and to simulate static fatigue due to an included strength parameter $\gamma$. The vast majority of asperity parameters have been already studied in previous inversion studies and are usually accessible from online databases. Dynamic deterministic modelling of aftershock series is still a challenge due to both the physical complexity and uncertainties related to the current state of the system. The FBM, on the other hand, produces similar statistical and fractal characteristics than real earthquake series, and its parameters can be regarded as analog to physical variables. Likewise, the FBM is able to simulate failure through static fatigue, creep failure or delayed rupture (Pradhan and Chakrabarti, 2003; Moreno et al., 2001).




**Figure 19.** A comparison between the data from table 5 and the results by TREMOLO for the events 7, 7a and 7b. (a) Maximum ruptured area, $A_{\mathrm{syn}}$; (b) mean maximum magnitude,$M_{\mathrm{syn}}$; (c) ratio of maximum event size $S_{\mathrm{syn}}$.

One disadvantage of TREMOL is that its output is highly dependent on the input which is based on information from kinematic models, and ,therefore, contain inherent uncertainties from inversion studies (see Table 5). TREMOL may be able to compensate some errors, but how far this possibility can be exploited needs to be investigated in the future. Further steps in the advance of the model just started to analyze a machine learning approach (Monterrubio-Velasco et al., 2018) which will exploit all the possibilities of this technique.

There are still issues that would likely be attacked in future tests:







**Figure 20.** Estimation of the characteristics of a future earthquake in the Guerrero seismic gap. (a) estimated rupture area, (b) rupture duration, (c) average of the mean magnitude considering three scale relations Eqs. 7, 8, and 9.

1. For our validations, we used earthquakes for which a suitable amount of information is available. How can the technique be applied to other events where only few information is available through, for instance, far field recordings of seismicity?



2. For our validation study, we used a simplified geometry of the real complex asperity geometries. However, other irregular asperity geometries may be introduced in future works.

3. The FBM is a pure statistical model and therefore gives only hints about underlying physical processes. So far, it does not take into consideration physical effects such a pore fluid pressure, soil amplification, stress relaxation of the upper mantle, reactivation of existing faults, volcanic activity or many more. One strength of the FBM is that an endless number of information layers can be included into the model which would allow to include physical properties and topography as well.

4. As it currently stands, TREMOL is not able to simulate complete seismic cycles. Rate-and-state friction models such as by Lapusta et al. (2000); Lapusta and Rice (2003) have the ability to reload stress. TREMOL is still in an early stage of development, and thus lacks a reloading feature.

Additional setbacks of TREMOL are that (1) the number of time-steps needs to be adjusted manually for every grid resolution and case scenario, and (2) it is based on a sequential algorithm. In order to save the stress history within every cell of the system, a consecutive algorithm is necessary which changes the state of the system with every time-step. This limits the integration of a parallel domain, but a parallel distributed memory is a good approach to solve the problem of large domains. As a result high performance computing facilities are required when very large grid sizes are used (Monterrubio-Velasco et al., 2018).

Overall, the results of TREMOL are promising. However, the results also point out the need for further modifications of the algorithm, and more intensive studies. Likewise, many questions are still left to be answered due to the model's early development stage. In the very near future, however, TREMOL may be a true alternative to classical approaches in seismology. The simple integration of layers of information makes TREMOL a simple model which can be easily modified to simulate the most complex scenarios. At the moment, TREMOL cannot compete with state-of-the-art and widely accepted rate-and-state friction based models, but it is a totally different, complementary, and promising approach which can provide important insights of earthquake physics and hazard assessment from a completely different perspective. The development of TREMOL and similar models should be therefore strongly encouraged and supported.

## 7 Conclusions

In this study, we present a FBM-based computer code called *stochasTic Rupture Earthquake MOdeL*, TREMOL, in order to investigate the rupture process of seismic asperities. We show that the model is capable of reproducing the main characteristics observed in real scenarios by means of few imput parameters. We carried out a parametric study in order to determine the optimal values for the three most important initial input parameters:

- $\pi_{asp}$: As long as the fault plane has a conservation parameter of $\pi_{bkg} = 0.67$, the conservation parameter of the asperity must be $\pi_{asp} \geq 0.76$ to ensure a realistic maximum rupture area.





- $\gamma_{\mathrm{asp}}$: The best strength interval for the asperity is $5 \pm 1 < \gamma_{\mathrm{asp}} < 14 \pm 1$. However, due to computational costs it is recommended to use the lowest value of $\gamma_{\mathrm{asp}} = 5 \pm 1$, since the number of necessary time-steps to activate the whole asperity increases strongly with the applied asperity strength (see Algorithm 1).

- $S_{\mathrm{a-Asp}}$: The generated magnitude can be controlled by parameter $S_{\mathrm{a-Asp}}$. This parameter is dependent on the earthquake of interest, and follows results from inversion studies data.

We also carried out a validation study employing 10 subduction earthquakes which occurred in Mexico. TREMOL proved that it is able to reproduce those scenarios with an appropriate tolerance.

A big advantage of our algorithm is the low number of free parameters ($L_{\mathrm{eff}}$, $W_{\mathrm{eff}}$, and $S_{\mathbf{a}}$) to obtain an appropriate rupture area, and magnitude assessment. Our code TREMOL allows its users to investigate the role of the initial stress configuration, and the material properties over the seismic asperity rupture. Both characteristics are key factors for understanding earthquake dynamics. The strengths of our FBM model are the simplicity in their implementation, the flexibility, and capability to model different rupture scenarios (*i.e.* asperity configurations) with varying mechanical properties within the asperities, and/or background area or fault plane. Although we simplified the expected complex asperity geometries, irregular asperity geometries may be introduced in future works. Another advantage is the analysis of earthquake dynamics from the point of view of deformable materials that break under critical stress. The results of TREMOL are promising. However, various assumptions and simplifications require further experiments and modifications of the algorithm to cover various tectonic settings. Likewise, the machine learning application by Monterrubio-Velasco *et al.* (2018) needs to be incorporated into the model to determine the optimal parameter ranges for different fault types and tectonic regimes. Although many questions are still left to be answered due to the model's early development stage, TREMOL proved to be a powerful tool which can deliver promising new insights into earthquake triggering processes. Our future work will investigate complex asperity configurations, earthquake doublets and stress transfer in three-dimensional domains.

*Code availability.* TREMOL

The TREMOL code is freely available at home page (https://zenodo.org/record/2079347.XAvuznWnFhF),from its GitHub repository (https://github.com/monterrubio-velasco), or by requesting the author (marisol.monterrubio@bsc.es, marisolmonterrub@gmail.com). In all cases, the code is supplied in a manner to ease the immediate execution under Linux platforms. User's manual documentation are provided in the archive as well.

*Acknowledgements.* M.Monterrubio-Velasco and A. Aguilar-Meléndez thank CONACYT for funding this research project. Quetzalcoatl Rodríguez-Pérez was supported by the Mexican National Council for Science and Technology (CONACYT) (Catedras program- project 1126). This project has received funding from the European Union's Horizon 2020 research and innovation programme under the Marie





Skłodowska-Curie grant agreement No 777778 MATHROCKS and from the Spanish Ministry Project TIN2016-80957-P. Initial funding for the project through grant UNAM-PAPIIT IN108115 is also gratefully acknowledged.





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
