# Peer review of "TREMOL: A stochastic rupture earthquake code based on the fiber bundle model. Application to Mexican subduction earthquakes."

_Geoscientific Model Development, 2018_

## Short Comment (SC1) · 28 Jan 2019

Dear authors,

in my role as Executive editor of GMD, I would like to bring to your attention our Editorial version 1.1:

http://www.geosci-model-dev.net/8/3487/2015/gmd-8-3487-2015.html

This highlights some requirements of papers published in GMD, which is also available

on the GMD website in the 'Manuscript Types' section:

http://www.geoscientific-model-development.net/submission/manuscript_types.html

In particular, please note that for your paper, the following requirement has not been met in the Discussions paper:

- "The main paper must give the model name and version number (or other unique identifier) in the title."

Please add a version number for TREMOL in the title upon your revised submission to GMD.

Yours,

Astrid Kerkweg

---

## Short Comment (SC2) · 29 Jan 2019

[revised manuscript text omitted]

**5.2 Model validation**

**5.2.1 Modelling 10 Mexican subduction zone earthquakes**

Based on the observations described in the previous section, we used $\gamma_{\text{asp}} = 5 \pm 1$, and $\pi_{\text{asp}} = 0.90$ in order to validate the model. We chose $\gamma_{\text{asp}} = 5 \pm 1$ because it represents the strength interval of $5 \pm 1 \leq \gamma_{\text{asp}} \leq 14 \pm 1$ with less computational costs.

5    We chose $\pi_{\text{asp}} = 0.90$ because it represents the relatively constant magnitude for the parameter range $0.76 \leq \pi_{\text{asp}} \leq 0.90$. In addition, $\pi_{\text{asp}} = 0.90$ enables to obtain the best approximation to the ratios of $S_{\text{a-Asp}}$. Both parameter choices ensure an appropriate reproduction of the asperity rupture area, the maximum magnitude and least computational payload.

Fig. 14 depicts a comparison between the (real) asperity area $A_{\text{real}}$ (Table 5), and the area of the largest simulated earthquake, $A_{\text{syn}}$. We plot the mean (blue dots), the minimum (green triangles), and the maximum (red triangles) of all 50 realizations for

10   each real earthquake event. Black squares represent the real asperity size. The results in Fig. 14 point out that $A_{\text{syn}}$ is almost identical to $A_{\text{real}}$ from Table 5 for the majority of earthquakes. Only three events show significant differences between synthetic and realistic maximum rupture area. Even in these cases, however, $A_{\text{real}}$ is located within the upper and lower limit of $A_{\text{syn}}$.

[Figure]

**Figure 10.** Mean and standard deviation of the maximum magnitude over 50 realizations depending on $\pi_{\mathrm{asp}}$.

Fig. 15 shows the statistical results of the synthetic maximum magnitude, $M_{\mathrm{syn}}$, determined for all 10 events. The real magnitudes from Table 5 are given as red markers. Black circles indicate the mean of $M_{\mathrm{syn}}$ of 50 realizations using Eq. 7, whereas blue and green markers indicate the magnitude following the equations 8 and 9, respectively. The error bars represent the standard deviation. We observed that the statistical parameters computed with TREMOL fit the magnitudes shown in Table

5   5. However, the computed magnitudes depend on the scale relation employed (Eq. 7, Eq. 8, and Eq. 9). Fig. 16 includes the mean of the three scale relations. Overall, the mean magnitude $\overline{M}_{\mathrm{syn}}$ and the expected magnitude $M_w$ show similar values. Given that the difference between the mean and the expected value (Table 5) is lower than $\Delta M_w < 0.5$ for the 10 events, we can affirm that the results of assessing the magnitude by means of TREMOL using a randomly modified asperity size, $S_{\mathrm{a-Asp}}$ (Eq. 17), are reasonable.

10   Fig. 17 shows the real ratio size $S_{\mathrm{a}}$ from table 5 (black squares) in comparison to the mean of the largest simulated earthquake, $S_{\mathrm{syn}}$ (blue squares). The standard deviation is represented as error bars. The results indicate that in most of the cases the computed $S_{\mathrm{syn}}$ range fits the expected $S_{\mathrm{a}}$ well. Note that for the Events 3, 7, and 8 the mean values are lower than the reported $S_{\mathrm{a}}$, while $S_{\mathrm{syn}}$ is overestimated for Events 2, 5 and 9. For Events 1, 4, 6 and 10 the estimated value of $S_{\mathrm{syn}}$ coincides with the expected one. However, the error bars enclose the expected values in all cases (Fig. 17). Moreover, if we compare Fig. 17

[Figure]

**Figure 11.** Mean and standard deviation of the ratio $S_{\text{syn}} = A_{\text{syn}}/A_{\text{eff}}$ over 50 realizations for different values of $\pi_{\text{asp}}$. The red line indicates the asperity ratio $S_{\text{a}}$ computed for event 7 (Table 5).

with Fig. 11 we observe that the employed strategy of randomly increasing asperity size (using Eq. 17) generate rupture areas similar to the ones proposed by Rodríguez-Pérez et al. (2018).

We also computed an equivalent rupture duration, $D_{\text{Aval}}$ using the equation proposed by Geller (1976) to calculate the rise time (Eq. 13 and 14). Rodríguez-Pérez and Ottemöller (2013) determined the rupture velocity $V_{\text{r}}$ (Ev. 3-8), which is a useful

5 parameter in order to validate our results. Fig. 18 shows the results of this analysis. In red we plot the values $V_{\text{r}}$ calculated by Rodríguez-Pérez and Ottemöller (2013), and in blue the $D_{\text{Aval}}$ based on Eq. 13 with $V_{\text{r}}$ provided by Table 5. The equivalent $D_{\text{
[revised manuscript text omitted]

---

## Referee Comment (RC1) · Anonymous Referee #1 · 5 Mar 2019

I think this is an interesting paper that introduces a new model to simulate the rupture process of asperities using a relatively low number number of parameters. The model is currently at an initial stage of development and the authors are working on improvements/extensions. This is ackwoneledged by the authors in the final sections of the paper.

My specific comments and/or observations about the paper are the following:

My major concern is the degree of uncertainties introduced by the initial choice of parameters, the fault parameters and whether this range of parameters holds to different tectonic settings . Fault length and width calculations are not trivial. I am not sure about the simplicity of the model, it seems to be dependent on studies for fault dimensions and for the model to be used it for another region, if my understanding is correct, the whole procedure (including the sensitivity analysis etc) has to be repeated.

In page 5 line 5 instead of the parenthesis could you give some examples?

In page 5 line 26 why uniform?

In page 6 line , how do you assign $\gamma$ref ? Would different values give you significantly different results in your final output?

In page 7 Why 0.98 and 0.02? Is there a reference? What criteria are used for this choice?

Are the range of values found in the sensitivity analysis unique to the examples in Mexico?

It would be interesting to see how your estimations compare with empirical relationships like Wells and Coppersmith (1994)

Some typos/syntax errors observed throughout the paper Eg. Page 1 line 15 should be earthquake magnitudes, Long sentence page 2 line 20 to 25 Page 5 line 11 should be: are computed in the... Figures are very far from the page they are referenced especially towards the end of the paper eg page 26 There is more than one section named Model validation

---

## Author Comment (AC1) · 14 Mar 2019

Response to comments by the reviewers and editor

We appreciate the comments and suggestions from the editor and reviewers, which have allowed us to greatly improve our manuscript. All comments have been addressed. In what follows we give response to the individual points raised.

Please also note the supplement to this comment:

[Figure]

https://www.geosci-model-dev-discuss.net/gmd-2018-323/gmd-2018-323-AC1-supplement.pdf

**Supplement:**

**Response to comments by the reviewers and editor**

We appreciate the comments and suggestions from the editor and reviewers, which have allowed us to greatly improve our manuscript. All comments have been addressed. In what follows we give response to the individual points raised.

**Reply to the comments of the Anonymous Referee #1.**

The following response format will be used:

- Question/Comment (from the reviewer)
- Answer (reply from the authors)
- Changes (new/modified text added to the manuscript)

###################################################################################
##########

**Question / comment**

"My major concern is the degree of uncertainties introduced by the initial choice of parameters, the fault parameters and whether this range of parameters holds to different tectonic settings. Fault length and width calculations are not trivial. I am not sure about the simplicity of the model, it seems to be dependent on studies for fault dimensions and for the model to be used it for another region, if my understanding is correct, the whole procedure (including the sensitivity analysis etc) has to be repeated."

**Answer:**

TREMOL is based on the Fiber Bundle model which is a stochastic cellular automata type model. Our goal is to investigate the basic rules of the FBM, adding some extra assumptions to simulate the rupture of asperities of different scenarios. It is clear that uncertainties play an important role.

All initial parameters depend on the dimensionality of the model (e.g. 2D or 3D), and the tectonic setting itself. Although the stochastic nature of TREMOL does not allow direct relations between known physical values and the parameters of the model, the stochastic approach may enable new insight into earthquake physics, and the stress transfer process during earthquakes.

In the case of $\pi_{Bkg}$, the range of values come from the results found in Monterrubio-Velasco (2013), Moreno et al. (2001), Monterrubio-Velasco et al. (2017). In these works, $\pi_{Bkg}$ was the only parameter required to define the global load transfer-value. Based on $\pi_{Bkg}$ we investigate $\pi_{asp}$ and $\gamma_{asp}$ in the work reported in the present paper.

In the case of fault dimensions, TREMOL requires the length and width of the effective rupture area, and the length and width of the asperity area as input parameters to initialize the model. In order to test the code using finite-fault source models, we used the source parameters derived by Rodríguez-Pérez et al. (2018) for the Mexican subduction zone.

If a user wants to test TREMOL v0.1 for another tectonic region, it is not necessary to perform a sensitivity analysis for each new project. In fact, part of the objectives of the present and previous works, was to carry out sensitivity analyses in order to identify values that can be applied to new models. However, it is a indeed true that TREMOL requires some geometric data such as the length and width of a fault (with the inherent uncertainties), otherwise we would not be able to characterize the seismic source of a particular zone, or seismic series. Such data can be obtained either from specific studies that have determined the dimensions of the faults or, alternatively, from the available relationships between the area and the magnitude of an earthquake. In the latter case, the inferred data will be the magnitude, and then the probable dimensions are computed using the relationships mentioned. Additionally, it is convenient to mention that, albeit TREMOL can model the behavior of a major fault, it is also capable of modelling seismicity at regions with no prescribed fault systems or at areas between faults. Compared to other methodologies, TREMOL requires few data to study, and model complex rupture scenarios.

A sensitivity analysis might be performed to test the range of variations according to uncertainties in the input (geometric and tectonic) parameters, but for our purpose it was considered unnecessary since we were interested in the range of variations of the model parameters. TREMOL v0.1 is rather inexpensive in computing power, hence a parametric analysis similar to that used in this paper can be obtained in a few days using a personal computer (approximately two minutes per realization, including postprocessing and graphs generation). Future work will include a parallel version to reduce the computational time.

**Question / comment**

"In page 5 line 5 instead of the parenthesis could you give some examples? "

**Answer/Changes:**

- other parameters (load-transfer value $\pi$, strength value $\gamma$, initial load values $\sigma$, and load threshold $\sigma_{th}$).

**Question / comment**

"In page 5 line 26 why uniform?"

**Answer:**

The uniform distribution is used because it is a simple random distribution that assigns initial load values in a heterogeneous manner. Being uniform, all values have equal probabilities of being assigned but that does not mean that all values are equal. This assumption allows simulating the unknown initial load (or stress) field. Moreover, an important reason for selecting this distribution is because of the results of tests  comparing observed seismicity with the synthetic catalogs resulted. In Monterrubio-Velasco et al. (2017) a Gaussian distribution was tested to assign the initial load values. The results were unsuccessful because the observations and the synthetic catalogs were completely different. Moreover,

the number of free parameters increases the number of degrees of freedom to choose the best range of distribution variables.

**Question / comment**

"In page 6 line , how do you assign $\gamma_{ref}$ ? Would different values give you significantly different results in your final output?"

**Answer:**

The parameter $\gamma_{asp}$ quantifies the "hardness" of the asperity relative to the background material. The value $\gamma_{ref}$ = 2 indicates that the strength in the asperity is twice that in the background. The range of $\gamma_{ref}$ explored (2,5,7,9,11,14) is based on an experimental choose. However, Fig. 12 shows a remarkable result because the mean of the maximum magnitude of an event as a function of $\gamma$asp does not change much for values $\gamma_{ref}$ > 3.
As $\gamma_{asp}$ increases the simulation requires a larger number of iterations to break a cell in the asperity, thus implying a larger computational cost.
Our election ($\gamma_{ref}$ = 4) assures a "stable" maximum magnitude in the lowest computational time. To visualize this fact we show the magnitude, the computational time [seconds] and the steps required to activate the whole asperity, for one execution, as function of $\gamma_{ref}$.
In this sense, we considered that a value $\gamma_{ref}$ = 4, is adequate from the computational point of view and to assure the statistically similar values of maximum magnitude although this value increases.

[Figure]

[Figure]

[Figure]

**Question / comment**

In page 7 Why 0.98 and 0.02? Is there a reference? What criteria are used for this choice?

**Answer:**

This assumption is in agreement with what is expected for the maximum shear stress directions with respect to the main stress orientation which gives rise to both synthetic and antithetic faulting (e.g. Stein and Wysession, 2008). We could assign a different percentage, but the goal was to assign as much of the load as possible to orthogonal neighbors. Although it is possible to include these values as free parameters, it would require additional degrees of freedom, since in this work it was not the study object we assumed values which way successful results in Monterrubio-Velasco et al. (2017).

**Question / comment**

Are the range of values found in the sensitivity analysis unique to the examples in Mexico?

**Answer:**

No, the values of $\pi$, $\gamma$, and $\sigma$ are generic for any simulation of similar type of earthquakes. The particular data for the Mexico examples are the effective area size and the asperity area which come from the results of finite-fault models.

**Question / comment**

It would be interesting to see how your estimations compare with empirical relationships like Wells and Coppersmith (1994)

**Answer:**

We considered more appropriate to compare the relationships of Strasser et al (2010), and Blaser et al (2010), since both were developed for subduction earthquakes. In the following table, we add the magnitude values using these three relations (Wells and Coppersmith (1994), Strasser et al (2010) and Blaser et al (2010), using as Area the mean value ("Simulated Area [km²]") reported in the manuscript.

| | Real Magnitude | Simulated Area [km²] | Wells and Coppersmith (1994) | Strasser et al., (2010) | Blaser et al., (2010) |
|---|---|---|---|---|---|
| Ev.1 | 7 | 140.65 | 6.18 | 6.26 | 6.44 |
| Ev.2 | 8.1 | 6281.24 | 7.79 | 7.65 | 8.32 |
| Ev.3 | 6.8 | 325.30 | 6.53 | 6.57 | 6.85 |
| Ev.4 | 7.4 | 727.36 | 6.87 | 6.86 | 7.25 |
| Ev.5 | 8 | 3506.22 | 7.54 | 7.44 | 8.03 |
| Ev.6 | 6.7 | 77.15 | 5.92 | 6.04 | 6.14 |
| Ev.7 | 7.4 | 691.49 | 6.85 | 6.84 | 7.23 |
| Ev.8 | 6.5 | 93.00 | 6.00 | 6.11 | 6.24 |
| Ev.9 | 8.2 | 3596.07 | 7.55 | 7.45 | 8.04 |
| Ev.10 | 7.1 | 403.97 | 6.62 | 6.65 | 6.96 |

Because the empirical magnitude-area relations can result in different magnitude values (Fig. 14), we decided to compute not only the magnitude but also the area of the maximum simulated earthquake (Fig. 15 and 16). However, if the user wants to use another empirical relation it is possible to add it in the script (TREMOL_singlets/postprocessing/calcuMagniSpaceTimeSinglets.jl).
In the particular case of Wells and Coppersmith (1994) relation is given as,

Mw = 4.07+ (0.98 * log10(A)),

We have plotted the mean magnitude, the maximum magnitude and the minimum magnitude considering Wells and Coppersmith (1994) relation and the mean value of the maximum magnitude obtained in TREMOL v0.1.
Comparing this plot with Fig. 14 in the manuscript, the magnitude values obtained by using Wells and Coppersmith (1994), are lower than those computed using Eqs. 7, 8 and 9. It is worth noting that Eqs. 7, 8 and 9 were developed for Mexican subduction earthquakes Rodríguez-Pérez and Ottemöller (2013).

[Figure]

**Question / comment**

Some typos/syntax errors observed throughout the paper Eg. Page 1 line 15 should be earthquake magnitudes, Long sentence page 2 line 20 to 25 Page 5 line 11 should be: are computed in the. . . Figures are very far from the page they are referenced especially towards the end of the paper eg page 26 There is more than one section named Model validation

**Answer**

In the revised manuscript typos/syntax errors have been corrected.

---

## Referee Comment (RC2) · Anonymous Referee #2 · 2 Apr 2019

The authors introduce a discrete stochastic model of earthquake fault zones based on the so-called time dependent fiber bundle model (FBM). Starting from the classical FBM the authors enhance the model construction by several key elements which are essential to capture the physics of earthquake generation. The main outcome of the work is the simulation code itself which was tested and validated. The role of the most important parameters of the model were tested carefully by computer simulations than the model was applied to study 10 earthquake of Mexican subduction zones. The authors demonstrate that if detailed information is available about a fault system the

model can be calibrated to give agreement with field measurements. In such situations the model can also be used for forecasting purpose up to some extent. The manuscript is a valuable contribution to the field. I recommend publication after responding to the following questions and comments:

1. When presenting the background of the model construction, right in the abstract the authors write that FBM is a discrete element model. Discrete element model (DEM) is not a proper wording here. DEM has a well defined meaning in physics and engineering and according to the generally accepted definition of DEM, FBM is not a DEM. FBMs are stochastic discrete models of materials failure.

2. When presenting FBMs the authors write that FBM is a numerical approach ... Actually, FBMs can be solved analytically in the mean field limit so I think "numerical approach" is not a proper wording here.

3. To represent materials' randomness the authors use uniform distributions throughout the work. This is questionable even if relatively good agreement is obtained with field measurements. The authors elaborate on this aspect of the model construction.

4. The authors mention on Page 7 that they are able to simulate "materials weakness or fatigue". I think what they have in the model is "weakening" and not simply weakness.

5. Load transfer is realized through the quantity \pi(x,y). According to the model construction \pi is set such that it generates a localized load sharing in the system, meaning that only nearest and next nearest neigbors share the load of a failed element. However, measurements usually show that long range stress transfer may play a role in earthquake sequences. Do the authors claim that at least in the studied cases short range stress transfer dominated the behaviour of faults?

6. In the validation process the authors only considered subduction earthquakes. Do the authors claim that the model is only applicable to this type of earthquakes? This is an important point which should be clarified in the discussion.

---

## Author Comment (AC3) · 4 Apr 2019

We reply the comments realized by the Anonymous Referee 2. We grateful his questions and comments. We also attach the manuscript indicating in blue color the changes that we introduce in order to improve the document considering his comments.

Kind regards,

Marisol Monterrubio-Velasco et al.

[Figure]

Please also note the supplement to this comment:
https://www.geosci-model-dev-discuss.net/gmd-2018-323/gmd-2018-323-AC3-supplement.pdf
* * *
[Figure]

**Supplement:**

**Response to comments by the reviewers and editor**

We appreciate the comments and suggestions from the editor and reviewers, which have allowed us to greatly improve our manuscript. All comments have been addressed. In what follows we give response to the individual points raised.

**Reply to the comments of the Anonymous Referee #2.**

The following response format will be used:

- Question/Comment (from the reviewer)
- Answer (reply from the authors)
- Changes (new/modified text added to the manuscript)

####################################################################

1. **Question / comment**

When presenting the background of the model construction, right in the abstract the authors write that FBM is a discrete element model. Discrete element model (DEM) is not a proper wording here. DEM has a well defined meaning in physics and engineering and according to the generally accepted definition of DEM, FBM is not a DEM. FBMs are stochastic discrete models of materials failure.

1. **Answer:**

The definition of the Fiber Bundle model as Discrete Element Model were found in Biswas et al. 2015 "Statistical Physics of Fracture, Breakdown and Earthquake: Effects of disorder and heterogeneity, page 34".
The Discrete Element method has a different meaning as Referee 2 says, since it is a numerical technique that models the interaction between individual particles and boundaries to predict bulk solids behavior.
It is our understanding that conceptually the Fiber Bundle model is a Discrete Element model since it describes the interaction of individual elements as a part of a collective medium.
However to solve any ambiguity we are agree to define FBMs are stochastic discrete models of materials failure.

2. **Question / comment**

When presenting FBMs the authors write that FBM is a numerical approach ... Actually, FBMs can be solved analytically in the mean field limit so I think "numerical approach" is not a proper wording here

**2. Answer:**

It is indeed true that, the "Equal Load sharing rule" is a mean-field model and can be solved analytically. However, the other extreme is the "Local load sharing" where the analytical solution becomes difficult and the best approach to solve the FBM equations is by carrying out a numerical procedure. In summary, the Referee's observation is appropriated. Therefore, we will substitute "numerical approach" by "mathematical tool" on Page 3.

**3. Question / comment**

To represent materials' randomness the authors use uniform distributions throughout the work. This is questionable even if relatively good agreement is obtained with field measurements. The authors elaborate on this aspect of the model construction.

**3. Answer:**

In seismicity process simulations the lack of knowledge of some important features such as the initial stress distribution or the strength and material heterogeneities generates a wide spectrum of uncertainties.
Without loss of generality, we consider that one way to aboard this issue can be considering a simple distribution such as the uniform distribution. We find that the validity of this assumption is given by the comparison of the simulated results with real data. It is possible that other distributions might also give similar results. However, the intention of TREMOL v0.1 is to propose a model which can be used to assign values to the unknown properties mentioned before, including different distributions. So we encourage users to try other possibilities.

**4. Question / comment**

The authors mention on Page 7 that they are able to simulate "materials weakness or fatigue". I think what they have in the model is "weakening" and not simply weakness

**4. Answer:**

Yes, we agree that the correct model process is a weakening process. We corrected the wording in that respect on Page 7.

**5. Question / comment**

Load transfer is realized through the quantity \pi(x,y). According to the model construction \pi is set such that it generates a localized load sharing in the system, meaning that only nearest and next nearest neighbors share the load of a failed element. However, measurements usually show that long range stress transfer may play a role in earthquake sequences. Do the authors claim that at least in the studied cases short range stress transfer dominated the behaviour of faults?

**5. Answer:**

In our simulation short range interactions convert to long-range processes through the avalanche mechanism in TREMOL v0.1. As the Referee 2 pointed out, the most obvious (or explicit) interaction range is the short since after a rupture in a individual element the load share is local and this produces a load concentration in neighbors cells, promoting ruptures in local manner (short-range). However, the long-range is also captured in more implicit way. This is allowed for two reasons, 1) the randomness in the initial stress distribution and 2) the selection rule for the next failed cell in the model, these produces that two cells far apart can be activated in successive steps, in analogy to a long-range interaction.

**6. Question / comment**

In the validation process the authors only considered subduction earthquakes. Do the authors claim that the model is only applicable to this type of earthquakes? This is an important point which should be clarified in the discussion.

**6. Answer:**

TREMOL v0.1 was focused on the asperity rupture process in a Subduction escenario because of the following reasons.
1. Due to the expertise of some of the co-authors in the teleseismic fault slip inversion applied to subduction zone earthquakes (Q. Rodríguez-Pérez, 2013). This provided the first motivation to study the rupture of asperities from the point of view of the Fiber Bundle model.
2. Related with the previous point, the data required to initialize the model was available to test the application of our hypothesis to a real case.
3. In a subduction event the rupture plane could be modeled in two dimensional domain, so this facilitated our study for the development of TREMOL v0.1
However, after validating the capability of the model, constraining the input parameters and analyzing the results, we consider that the coceptual basis of TREMOL can be expanded to model other tectonic regimes. In fact, some work is currently carried out in this sense. We recently submitted the study of the interaction of Faults systems and the production of aftershocks (Monterrubio-Velasco et al. In Review se-2019-65. Modeling active fault systems and seismic events by using a Fiber Bundle model. Example case: Northridge aftershock sequence).
We are also working on a three dimensional version to simulate the mainshock and aftershocks (Sholz D. (2018). Numerical simulations of stress transfer as a future alternative to classical Coulomb stress changes?: Investigation of the El Mayor Cucapah event. Unpublished Master thesis).

---

## Author Response (AR2)

Dear Thomas Poulet,

Thank you for your comments. We include all your questions into the manuscript in order to improve it. In red color you will see the number of the question and in blue color our answer. As a part of the question 1.4 we also include a new figure Fig. 21 (pag. 36). Also we improve Fig. 1.

We grateful your attention.
Kind regards,

Marisol Monterrubio-Velasco et al.